# DISTRIBUTIONAL VISION LANGUAGE ALIGNMENT BY CAUCHY-SCHWARZ DIVERGENCE

**Wenzhe Yin**[1]* **Zehao Xiao**[1]* **Pan Zhou**[5]† **Shujian Yu**[3,4] **Jiayi Shen**[1]
**Jan-Jakob Sonke**[2] **Efstratios Gavves**[1]
[1]University of Amsterdam  [2]The Netherlands Cancer Institute  [3]Vrije Universiteit Amsterdam
[4]The Arctic University of Norway  [5]Singapore Management University

## ABSTRACT

Vision-language alignment is crucial for various downstream tasks such as cross-modal generation and retrieval. Previous multimodal approaches like CLIP utilize InfoNCE to maximize mutual information, primarily aligning pairwise samples across modalities while overlooking distributional differences. In addition, InfoNCE has inherent conflict in terms of alignment and uniformity in multimodality, leading to suboptimal alignment with modality gaps. To overcome the limitations, we propose CS-Aligner, a novel framework that performs distributional vision-language alignment by integrating Cauchy-Schwarz (CS) divergence with mutual information. CS-Aligner captures both the global distribution information of each modality and the pairwise semantic relationships. We find that the CS divergence seamlessly addresses the InfoNCE's alignment-uniformity conflict and serves complementary roles with InfoNCE, yielding tighter and more precise alignment. Moreover, by introducing distributional alignment, CS-Aligner enables incorporating additional information from unpaired data and token-level representations, enhancing flexible and fine-grained alignment in practice. Experiments on text-to-image generation and cross-modality retrieval tasks demonstrate the effectiveness of our method on vision-language alignment.

## 1 INTRODUCTION

Vision-language alignment aims to map the paired text and image inputs into a shared feature space, enabling success across diverse applications such as image-text retrieval (Huang et al., 2024; Koukounas et al., 2024) and text-to-image (T2I) generation (Ramesh et al., 2022; Razzhigaev et al., 2023). As a pioneering work in this field, CLIP (Radford et al., 2021) leverages InfoNCE loss (a.k.a. contrastive loss) to maximize the mutual information between paired text and image representations, effectively capturing pairwise and semantic relationships. Its versatility has made it a foundation for many multimodal tasks (Ramesh et al., 2022; Mokady et al., 2021).

Despite its success, CLIP and its variants (Zhai et al., 2023; Sun et al., 2023) exhibit a persistent modality gap, a misalignment between text and image representations in the shared latent space. As shown in Fig. 1a, text and image embeddings often fail to align precisely and may remain separated from each other. This phenomenon has been widely observed (Zhou et al., 2023; Liang et al., 2022; Shi et al., 2023) and is attributed to issues such as cone effects (Liang et al., 2022) or suboptimal latent space geometry (Shi et al., 2023). Intriguingly, Liang *et al.* (Liang et al., 2022) observed that CLIP's InfoNCE loss could inadvertently exacerbate the modality gap, since, as analyzed in Sec. 2, InfoNCE loss can be decomposed into alignment and uniformity components, which indeed conflict with each other during vision-language alignment.

Several strategies have been proposed to mitigate the modality gap, such as projection modules with cosine similarity (Zhou et al., 2023; Gao et al., 2024; Huang et al., 2024) and geodesic multimodal mixup (Oh et al., 2024). UnCLIP-based models like DALL-E 2 (Ramesh et al., 2022) employ text-to-image prior modules (e.g., diffusion models) to map text embeddings to image feature space

---

*Equal contribution.
†Correspondence to: Pan Zhou <panzhou@smu.edu.sg>.

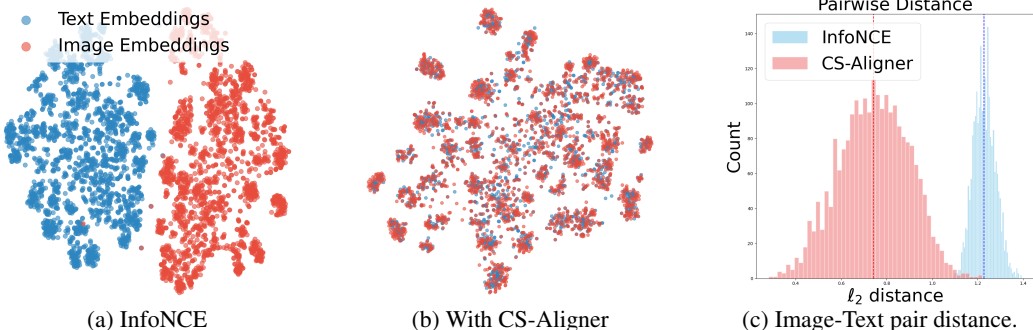

| (a) InfoNCE | (b) With CS-Aligner | (c) Image-Text pair distance. |

Figure 1: **TSNE visualizations of CLIP text and image features without (a) and with (b) CS-Aligner.** The original CLIP feature distributions reveal a clear domain gap (a). Adapting the model with our CS-Aligner effectively eliminates the modality gap, leading to tighter alignment (b). Consequently, CS-Aligner yields a lower overall $\ell_2$ distance between paired image-text features (c).

for alignment. A more recent alternative Eclipse (Patel et al., 2024) uses $\ell_2$ loss to train a prior adapter for text and image alignment. These works aim to transform representations across modalities for alignment. However, they explore alignment sample-wisely, heavily relying on pairwise data. Although sample-wise alignment effectively captures semantic information, it falls short in aligning entire data distributions. Similar to the InfoNCE in CLIP, the methods struggle to match the representation spaces across modalities, ultimately limiting the overall alignment. The reliance on carefully curated text-image pairs also limits the scalability and applicability to real-world scenarios with unpaired and noisy datasets (Lin et al., 2014; Li et al., 2023b). Moreover, the theoretical conflict of InfoNCE for vision-language alignment is still under exploration.

To address these challenges, we propose CS-Aligner, a novel distributional approach that incorporates Cauchy-Schwarz (CS) divergence (Principe et al., 2000b) for vision-language alignment. As a symmetric measure, CS divergence robustly and efficiently estimates the distance between any representation distributions without parametric distributional assumptions, making it highly suitable for multimodal distribution alignment. Furthermore, we analyze the alignment–uniformity conflict of InfoNCE in multimodal settings and show that CS divergence effectively mitigates it while remaining compatible with InfoNCE via kernel density estimation (KDE) (Parzen, 1962). This enables CS-Aligner to align vision–language representations at distributional and sample-wise levels, capturing global modality and local semantics, yielding more comprehensive, consistent, and tighter alignment as shown in Figs. 1b and 1c.

Moreover, the distributional nature of CS-Aligner enables alignment with unpaired multimodal data, including cases where a) a single image is associated with multiple captions, or b) vision and language inputs are entirely unpaired. This flexibility allows our method to leverage rich and unstructured datasets and improve alignment robustness beyond curated benchmarks. Beyond unpaired alignment, we introduce a token-level alignment strategy, which further enriches the multimodal representation by aligning fine-grained visual and textual tokens, enhancing the semantic precision of the learned embeddings. Extensive experiments on downstream tasks, including T2I generation and image-text retrieval, demonstrate the effectiveness of our approach.

## 2 InfoNCE is insufficient for alignment

Previous multimodal methods (for vision-language) like CLIP (Radford et al., 2021) learn text and image representations in a shared space by maximizing lower bounds (e.g., InfoNCE (Oord et al., 2018)) of mutual information between modalities:

$$I(\mathbf{x}; \mathbf{y}) = \int \int p(\mathbf{x}, \mathbf{y}) \log \frac{p(\mathbf{x}, \mathbf{y})}{p(\mathbf{x})p(\mathbf{y})} \, d\mathbf{x} \, d\mathbf{y}, \tag{1}$$

where $p(\mathbf{x})$ and $p(\mathbf{y})$ are respectively the distributions of image and text features, and $p(\mathbf{x}, \mathbf{y})$ denotes their joint probability. Although widely used, it suffers from two limitations.

**Limitation1: Mutual information is insufficient for multimodal alignment.** Although widely adopted, mutual information alone is insufficient for effective modality alignment (Liang et al.,

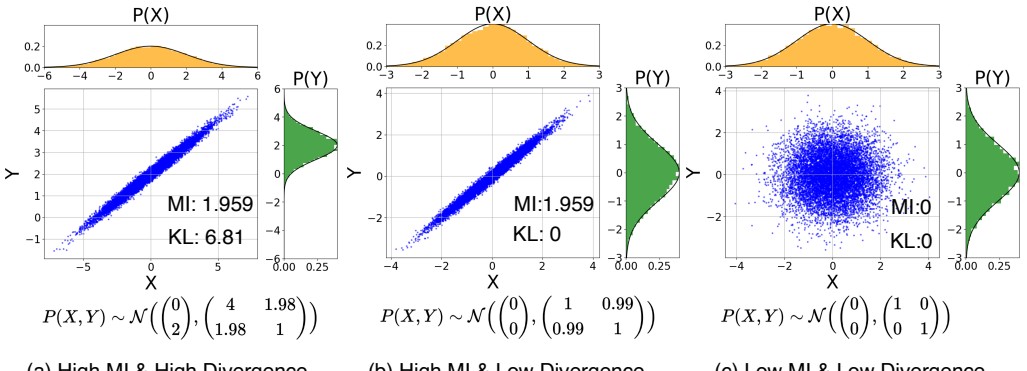

Figure 2: **Toy examples: mutual information (MI ↑) and distribution divergence (↓) between two distributions.** Distributions with the same high mutual information value can exhibit either large (a) or small (b) distributional distances, demonstrating that MI alone is insufficient for multimodal alignment. Moreover, distribution divergence measures the closeness between distributions but does not guarantee that the underlying random variables are statistically correlated (c).

2022). The reason is that mutual information quantifies the statistical dependence between two random variables (Cover, 1999), ensuring correlation maximization between two random variables. However, it does not guarantee that the distributions $p(\mathbf{x})$ and $p(\mathbf{y})$ are statistically similar or close to each other in terms of their underlying distributions. In other words, the embedding distributions of two modalities can differ significantly or be far apart, yet exhibit strong dependence.

We illustrate this issue using a toy example in Fig. 2. Fig. 2a shows that despite strong dependence and high mutual information, the representation distributions of two representations or random variables can remain misaligned and be far from each other, resulting in a high divergence. This issue is also observed in the CLIP model pretrained with InfoNCE, where the vision and language representations exhibit a noticeable distributional gap, as shown in Fig. 1a. This gap results in inconsistently aligned multimodal features, hindering the clear representation of shared semantics and disrupting effective mapping between modalities. Ultimately, this misalignment degrades performance in downstream tasks, including cross-modality generation. Ideally, the desired multimodal representations should be highly correlated with low distributional divergence, as depicted in Fig. 2b. Notably, although directly minimizing the divergence between distributions may reduce the distributional gap, it risks creating independent multimodal distributions without common semantic information (Fig. 2c). Therefore, maximizing mutual information and minimizing divergence complement each other to achieve effective multimodal representation alignment. Details are provided in Appendix A.

**Limitation2: InfoNCE includes conflicting terms for multimodal alignment.** In practice, mutual information is often optimized via the InfoNCE loss (Oord et al., 2018) which estimates $I(\mathbf{x}; \mathbf{y})$ using paired image-text data $\{(\mathbf{x}_i, \mathbf{y}_i)\}_{i=1}^N$ and contains image-text and text-image alignment terms:

$$\mathcal{L}_{\text{InfoNCE}} = -\frac{1}{2N} \sum_{i=1}^N \left( h(\mathbf{x}_i, \mathbf{y}_i) + h(\mathbf{y}_i, \mathbf{x}_i) \right), \qquad h(\mathbf{x}, \mathbf{y}) = \log \frac{\exp\left(\text{sim}(\mathbf{x}, \mathbf{y})/\tau\right)}{\sum_{j=1}^N \exp\left(\text{sim}(\mathbf{x}, \mathbf{y}_j)/\tau\right)}, \tag{2}$$

where $\text{sim}(\cdot, \cdot)$ is cosine similarity and $\tau$ is temperature. Critically, the InfoNCE loss in Eq. (2) requires paired data $\{(\mathbf{x}_i, \mathbf{y}_i)\}_i^N$, and cannot work under unpaired setting.

As analyzed in Wang & Isola (2020), the InfoNCE loss can be decomposed as the sum of the alignment ($\mathcal{L}_{\text{align}}$) and uniformity ($\mathcal{L}_{\text{uniform}}$) terms i.e., $\mathcal{L}_{\text{InfoNCE}} \approx \mathcal{L}_{\text{align}} + \mathcal{L}_{\text{uniform}}$:

$$\mathcal{L}_{\text{align}} \triangleq \mathbb{E}_{(\mathbf{x}, \mathbf{y}) \sim p_{\text{pair}}} \left[ ||\mathbf{x} - y||_2^\alpha \right], \quad \mathcal{L}_{\text{uniform}} \triangleq \log \mathbb{E}_{\mathbf{x}, \mathbf{y} \overset{\text{i.i.d.}}{\sim} p(\mathbf{x}, \mathbf{y})} \left[ \exp(-t ||\mathbf{x} - \mathbf{y}||_2^2) \right], \tag{3}$$

where $t$ and $\alpha$ are hyperparameters. $p_{\text{pair}}$ denotes the image-text pairs distribution. Minimizing $\mathcal{L}_{\text{align}}$ encourages pairwise alignment. In unimodality, minimizing $\mathcal{L}_{\text{uniform}}$ promotes representations that are uniformly distributed on the unit hypersphere, a desirable property for representation learning (Wang & Isola, 2020). However, in multimodal alignment, $\mathcal{L}_{\text{uniform}}$ may conflict with $\mathcal{L}_{\text{align}}$.

***Remark*** 2.1. The uniformity and alignment terms in InfoNCE conflict with each other in multimodal alignment. Applying Taylor expansions ($\mathbb{E}(e^{-\mathbf{x}}) \approx 1 - \mathbb{E}(\mathbf{x})$ and $\log(1 - \mathbf{x}) \approx -\mathbf{x}$) on $\mathcal{L}_{\text{uniform}}$,

the uniformity term becomes:

$$\mathcal{L}_{\text{uniform}} \approx -t\mathbb{E}_{\mathbf{x},\mathbf{y}\sim p(\mathbf{x},\mathbf{y})}\left[||\mathbf{x}-\mathbf{y}||_2^2\right] = -t\mathbb{E}_{(\mathbf{x},\mathbf{y})\sim p_{\text{pair}}+p_{\text{unpair}}}\left[||\mathbf{x}-\mathbf{y}||_2^2\right], \tag{4}$$

where $p(\mathbf{x},\mathbf{y}) = p_{\text{pair}} + p_{\text{unpair}}$, and $p_{\text{unpair}}$ denotes the distribution of unpaired image and text. Consequently, the combination of the two (InfoNCE) can be written as:

$$\mathcal{L}_{\text{align}} + \mathcal{L}_{\text{uniform}} \approx \mathbb{E}_{(x,y)\sim p_{\text{pair}}}\left[||\mathbf{x}-\mathbf{y}||_2^\alpha\right] - t\mathbb{E}_{(\mathbf{x},\mathbf{y})\sim p_{\text{pair}}+p_{\text{unpair}}}\left[||\mathbf{x}-\mathbf{y}||_2^2\right]. \tag{5}$$

The alignment contribution ($\mathcal{L}_{\text{align}}$) in Eq. (3) can be largely suppressed or even canceled (when $t = 1$) due to the opposing term in Eq. (5), leaving only negative pairs influential. Essentially, $\mathcal{L}_{\text{align}}$ promotes alignment across modalities, whereas $\mathcal{L}_{\text{uniform}}$ encourages dissimilarity among negative pairs *without preserving intra-modal structure*. This inherent conflict can result in local minima, driving alignment and uniformity in opposing directions and ultimately leading to a modality gap. Thus, InfoNCE alone may lead to suboptimal alignment between modalities.

## 3 METHODOLOGY

In this section, we address the incapability of mutual information on aligning distributions and the conflicts in InfoNCE for multimodal alignment. To this end, we first introduce a novel distributional multimodal alignment framework, CS-Aligner. Then, we analyze that with the KDE, the proposed method is able to address the uniformity-alignment conflicts of InfoNCE. Finally, we extend CS-Aligner to the unpaired data, including token-level alignment.

### 3.1 CS-ALIGNER: DISTRIBUTIONAL MULTIMODAL ALIGNMENT

To mitigate limitation 1 in Sec. 2, we explicitly minimize the distribution divergence between $p(\mathbf{x})$ and $p(\mathbf{y})$. In practice, $p(\mathbf{x})$ and $p(\mathbf{y})$ may follow arbitrary distributions with minimal intersection, which may often occur in the multimodal setting. Hence, a robust divergence metric must accommodate unpredictable variability and limited support overlap for effective distribution alignment.

To this end, we propose a distributional alignment framework, namely ***CS-Aligner***, which leverages the CS divergence ($D_{\text{CS}}$), as illustrated in Fig. 3. The objective is:

$$\min -I(\mathbf{x};\mathbf{y}) + \lambda D_{\text{CS}}(p(\mathbf{x}), p(\mathbf{y})), \tag{6}$$

where $\lambda$ is a hyperparameter balancing the mutual information term and the divergence penalty. CS divergence, $D_{\text{CS}}$, is a symmetric and robust metric to quantify the distance between any two probability density functions $p$ and $q$, defined over the same support $\omega$ as:

$$D_{\text{CS}}(p;q) = -\log\left(\left(\int p(\omega)q(\omega)d\omega\right)^2 \Big/ \left(\int p(\omega)^2 d\omega \int q(\omega)^2 d\omega\right)\right), \tag{7}$$

The CS divergence satisfies $0 \le D_{\text{CS}} < \infty$, and equals zero if and only if $p = q$. By introducing $D_{\text{CS}}$ in Eq. (6), instead of solely minimizing pairwise distance, our method also aligns the distributions of modalities, leading to more robust and efficient multimodal alignment, as shown in Fig. 3.

**CS divergence estimation.** To estimate CS divergence, we introduce non-parametrical KDE. The non-parametric KDE means that it does not assume any specific parametric form for the underlying distribution. This eliminates the need for explicit parametric assumptions about the underlying distributions. This provides significant flexibility in measuring distributional distance. Given *i.i.d.* samples $\{\mathbf{x}_i\}_{i=1}^M \sim p(\mathbf{x})$ and $\{\mathbf{y}_i\}_{i=1}^N \sim p(\mathbf{y})$, the empirical CS divergence estimator is given by Jenssen et al. (2006):

$$\widehat{D}_{\text{CS}}(p(\mathbf{x});p(\mathbf{y})) = \log\left(\frac{1}{M^2}\sum_{i,j=1}^M \kappa(\mathbf{x}_i,\mathbf{x}_j)\right) + \log\left(\frac{1}{N^2}\sum_{i,j=1}^N \kappa(\mathbf{y}_i,\mathbf{y}_j)\right) - 2\log\left(\frac{1}{MN}\sum_{i=1}^M\sum_{j=1}^N \kappa(\mathbf{x}_i,\mathbf{y}_j)\right). \tag{8}$$

where $\kappa$ is a kernel function such as Gaussian $\kappa_\sigma(\mathbf{x},\mathbf{y}) = \exp(-\|\mathbf{x}-\mathbf{y}\|_2^2/2\sigma^2)$ with kernel width $\sigma$. This estimator is symmetric, differentiable, and computationally efficient, making it suitable for multimodal alignment. Moreover, the third term in Eq. (8) ensures that $\widehat{D}_{\text{CS}}(p(\mathbf{x});p(\mathbf{y})) \to \infty$ only

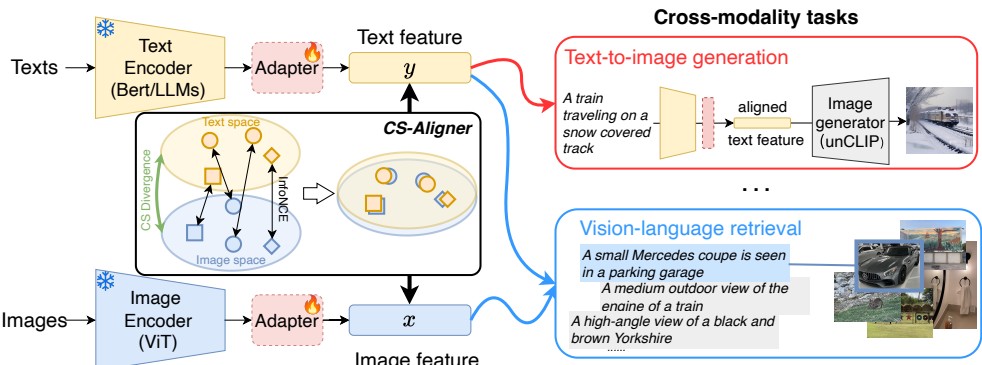

Figure 3: **Illustration of CS-Aligner.** We achieve vision-language alignment by freezing the pre-trained text and image encoders and applying parameter-efficient fine-tuning methods (e.g., adapter) with our CS-Aligner. CS-Aligner optimizes the adapters using the aggregated CS divergence and In-foNCE, as formulated in Eq. (6). Once aligned, the adapters are utilized for various cross-modality tasks: the aligned text adapter facilitates text-to-image generation without additional modifications, while the aligned multimodal adapters are used for vision-language retrieval.

when $\mathbb{E}(\kappa(\mathbf{x}, \mathbf{y})) \to 0$ (i.e., when the distributions do not overlap). However, as long as there is a nonzero overlap between the distributions, the estimator remains well-defined and valid.

Hence, CS-Aligner remains reliable even when the two distributions initially have limited overlap, a common scenario in multimodal tasks. Additionally, its symmetry and non-parametric estimation properties ensure consistent and unbiased multimodal alignment. Consequently, our method ensures both semantic and distributional alignment, enabling robust and efficient multimodal learning.

When estimating the mutual information $I(\mathbf{x}, \mathbf{y})$ via InfoNCE (Eq. (2)), unlike other distribution divergences, CS divergence effectively addresses InfoNCE's inherent alignment-uniformity conflict.

**Uniformity and Alignment with CS Divergence.** Using the Gaussian kernel $\kappa_t(\mathbf{x}, \mathbf{y}) = \exp(-t\|\mathbf{x} - \mathbf{y}\|_2^2)$ for CS divergence and combining the alignment and uniformity components of InfoNCE, the full objective of Eq. (6) can be expressed as

$$\mathcal{L} = \mathbb{E}_{(\mathbf{x},\mathbf{y}) \sim p_{\text{pos}}} \left[ \|\mathbf{x} - \mathbf{y}\|_2^\alpha \right] + \log \mathbb{E}_{\mathbf{x} \sim \mathbf{p}(\mathbf{x}), \mathbf{y} \sim \mathbf{p}(\mathbf{y})} \left[ \kappa_t(\mathbf{x}, \mathbf{y}) \right]$$
$$+ \lambda \Big( \log \mathbb{E}_{\mathbf{x},\mathbf{x}' \sim p(\mathbf{x})} \left[ \kappa_t(\mathbf{x}, \mathbf{x}') \right] + \log \mathbb{E}_{\mathbf{y},\mathbf{y}' \sim p(\mathbf{y})} \left[ \kappa_t(\mathbf{y}, \mathbf{y}') \right] - 2 \log \mathbb{E}_{\mathbf{x} \sim p(\mathbf{x}), \mathbf{y} \sim p(\mathbf{y})} \left[ \kappa_t(\mathbf{x}, \mathbf{y}) \right] \Big).$$
$$(9)$$

When $\lambda = 1$, this reduces to the following alignment–uniformity decomposition:

$$\mathcal{L} = \underbrace{\mathbb{E}_{(\mathbf{x},\mathbf{y}) \sim p_{\text{pair}}} \left[ \|\mathbf{x} - \mathbf{y}\|_2^\alpha \right] - \log \mathbb{E}_{\mathbf{x} \sim p(\mathbf{x}), \mathbf{y} \sim p(\mathbf{y})} \left[ \exp\left(-t\|\mathbf{x} - \mathbf{y}\|^2\right) \right]}_{\text{Alignment}}$$
$$+ \underbrace{\log \mathbb{E}_{\mathbf{x},\mathbf{x}' \sim p(\mathbf{x})} \left[ \exp\left(-t\|\mathbf{x} - \mathbf{x}'\|^2\right) \right]}_{\text{Uniformity on } \mathbf{x}} + \underbrace{\log \mathbb{E}_{\mathbf{y},\mathbf{y}' \sim p(\mathbf{y})} \left[ \exp\left(-t\|\mathbf{y} - \mathbf{y}'\|^2\right) \right]}_{\text{Uniformity on } \mathbf{y}}. \quad (10)$$

***Remark*** 3.1. For the alignment part, CS-Aligner promotes both the matching of image-text pairs and the alignment of global distributions. For uniformity, CS-Aligner encourages dispersion within each modality independently, rather than across modalities, which could otherwise conflict with the alignment objective. Thus, our method simultaneously fosters both alignment and uniformity while avoiding the potential conflicts inherent in InfoNCE.

***Remark*** 3.2. The connection between CS divergence and InfoNCE becomes evident when analyzing both terms from a cosine similarity perspective. For a characteristic kernel $\kappa(\mathbf{x}, \mathbf{y}) = \langle \phi(\mathbf{x}), \phi(\mathbf{y}) \rangle_{\mathcal{H}}$, where $\phi$ maps samples to a Reproducing Kernel Hilbert Space (RKHS) $\mathcal{H}$, the mean embeddings are: $\boldsymbol{\mu}_x = \frac{1}{m} \sum_{i=1}^m \phi(\mathbf{x}_i)$ and $\boldsymbol{\mu}_y = \frac{1}{n} \sum_{i=1}^n \phi(\mathbf{y}_i)$, The CS divergence can then be expressed in a form that evaluates the cosine similarity between distributions in RKHS:

$$\hat{D}_{\text{CS}}(p(\mathbf{x}); p(\mathbf{y})) = -2 \log \left( \frac{\langle \boldsymbol{\mu}_x, \boldsymbol{\mu}_y \rangle_{\mathcal{H}}}{\|\boldsymbol{\mu}_x\|_{\mathcal{H}} \|\boldsymbol{\mu}_y\|_{\mathcal{H}}} \right) = -2 \log \text{sim}(\boldsymbol{\mu}_x, \boldsymbol{\mu}_y), \quad (11)$$

Similarly, InfoNCE evaluates cosine similarity between paired samples (Eq. (2)). This dual-level similarity assessment underscores the synergy between CS divergence and mutual information, offering a unified and robust framework for multimodal alignment.

Therefore, CS divergence is compatible with InfoNCE and effectively addresses the inherent conflict between uniformity and alignment, a property not shared by other distribution distance metrics. Detailed comparisons with other metrics are provided in the Appendix D.

## 3.2 EXTEND CS-ALIGNER TO UNPAIRED DATA

Benefiting from the distributional alignment, we further propose extensions of CS-Aligner, which leverage additional information in unpaired data. While the mutual information estimation (InfoNCE) part requires pairwise data, the CS divergence estimator (Eq. (8)) can operate seamlessly on unpaired data without introducing additional computation. This unique capability enables CS-Aligner to extend beyond traditional pairwise multimodal alignment by incorporating additional distributional information from unpaired data or tokens. Below, we introduce two novel directions.

**Unpaired vision-language alignment.** Our method leverages two forms of unpaired alignments: (1) images with multiple captions, and (2) independently sampled unpaired images and texts. The unpaired alignments are achieved using Eq. (8), where $\{x_i\}_{i=1}^{M}$ and $\{y_j\}_{j=1}^{N}$ can be independent with $M \neq N$. In both scenarios, our method leverages more uncurated unpaired data for distributional multimodal alignment, providing greater flexibility and robustness.

**Vision-language token alignment.** We propose a novel intra-sample distribution alignment approach between vision and language tokens. Unlike CLIP-based models (Radford et al., 2021) aligning only the "CLS" tokens of vision and text, our method aligns all tokens for finer-grained alignment. Specifically, each vision feature $\mathbf{x}_i \in \mathbb{R}^{V \times D}$ is modeled as a token distribution $p(\mathbf{x}_i)$ containing $V$ vision tokens, while each text feature $\mathbf{y}_i \in \mathbb{R}^{L \times D}$ is represented as a token distribution $p(\mathbf{y}_i)$ with $L$ text tokens. $D$ denotes the feature dimension. We compute CS divergence between vision and text token distributions, and obtain an internal token-wise alignment loss:

$$\mathcal{L}_{\text{token}} = \frac{1}{B} \sum_{i=1}^{B} \widehat{D}_{\text{CS}}(p(\mathbf{x}_i); p(\mathbf{y}_i)), \tag{12}$$

where $B$ is the batch size. In general, $V \neq L$, and vision and language tokens do not have a direct pairing, making InfoNCE inapplicable for estimation. Through our distributional alignment, Eq. (12) enables comprehensive alignment across all tokens, capturing more details and potentially enhancing fine-grained alignment.

## 3.3 PARAMETER-EFFICIENT MULTIMODAL ALIGNMENT

We demonstrate the effectiveness of our CS-Aligner by performing vision-language alignment in a parameter-efficient manner using pretrained vision and language models, such as CLIP and large language models (LLMs) (Dubey et al., 2024). To adapt these pretrained models, we employ two widely used frameworks: adapter (Gao et al., 2024) and LoRA (Hu et al., 2021). The adapter and LoRA enable efficient alignment of the multimodal large-scale pretrained models, without requiring extensive computational resources. The whole framework is demonstrated in Fig. 3.

**Adapter & LoRA alignment.** We add a lightweight transformer (Vaswani, 2017) on top of the pretrained model as an adapter that projects text or image embeddings into a shared space; optionally, we can insert trainable low-rank (LoRA) matrices into the text encoder's weights to enable fine-grained adjustments, aligning the representations with the other modality.

## 4 EXPERIMENTS

We evaluate our method on two tasks to illustrate its vision-language alignment ability: text-to-image (T2I) generation in Section 4.1 and image-text retrieval in Section 4.2. Note that we focus on the vision-language alignment and use the generation task as a proxy to measure it. Additionally, we provide the image-text classification and the image captioning results in Appendix H. We also

present the computation complexity and stability analysis in Appendix E, and additional ablation studies in Appendix H.1.

## 4.1 TEXT TO IMAGE GENERATION

**Datasets.** Following a previous T2I approach (Patel et al., 2024), we train our method on four datasets: **MSCOCO** (Lin et al., 2014), **CC3M** (Sharma et al., 2018), **CC12M** (Changpinyo et al., 2021), and **LAION-HighResolution-5M** (Schuhmann et al., 2022). MSCOCO contains 80K images paired with multiple captions. CC3M and CC12M include about 2.5M and 10M image-text pairs, respectively. LAION-HighResolution comprises 175M high-resolution pairs, from which we select 5M for training. We evaluate the aligned model on the MSCOCO 30K validation set.

**Experimental setup.** We build our method based on unCLIP-style approaches (e.g., DALL-E-2 (Ramesh et al., 2022), Karlo (Donghoon et al., 2022), Kandinsky (Razzhigaev et al., 2023)). These methods train a diffusion prior module on large-scale datasets (hundreds of millions of samples) to map text into the image representation space, and use a decoder to generate images.

Differently, CS-Aligner trains an adapter to align text representations to image feature space on small-scale datasets, e.g., MSCOCO (0.08M), CC3M (3M), and CC12M (12M), and LAION-HighRes subset (5M). After alignment, we directly process the aligned text features using the pretrained decoder of the large-scale methods (e.g., Karlo and Kandinsky) to generate images, without additional prior modules or multiple diffusion steps. We evaluate generation quality with the FID score (Heusel et al., 2017), which measures how closely generated images match the real image distribution. This metric is particularly well-suited for evaluating modality alignment, as it directly reflects the distribution distance. Additional details can be found in Appendix G.

**Baselines.** Our baselines consists of both large-scale methods Karlo, Kandinsky, Wurstchen (Pernias et al., 2023), Stable Diffusion (Rombach et al., 2022) (SD v2.1 and SD-unClip), and the recent small-scale alignment method Eclipse. We also compare with the most recent multimodal alignment method (Almudévar et al., 2025) (denoted as IB) on the generation task. For fairness, we use the same Transformer adapter as Eclipse (also for (Almudévar et al., 2025)) and only align the "CLS" tokens, highlighting the advantages of our distributional alignment.

Table 1: **Comparisons with T2I methods.** Our method outperforms large-scale diffusion-based methods and the recent small-scale (alignment) methods (Eclipse and IB (Almudévar et al., 2025)).

| Methods | Datasize (M) | FID |
|---|---|---|
| **Large-scale methods** | | |
| SD v2.1 | 2000 | 14.51 |
| SD-unclip v2.1 | 2000 | 13.15 |
| Wurstchen | 1420 | 23.60 |
| DALL-E2 | 250 | 10.65 |
| Kandinsky | 177 | 20.48 |
| Karlo | 115 | 20.64 |
| **Small-scale alignment** | | |
| IB + Kandinsky decoder | $0.08_{(COCO)}$ | 150.52 |
| Eclipse + Kandinsky decoder | $0.08_{(COCO)}$ | 16.53 |
| Ours + Kandinsky decoder | $0.08_{(COCO)}$ | **12.62** |
| Eclipse + Karlo decoder | $0.08_{(COCO)}$ | 23.67 |
| Ours + Karlo decoder | $0.08_{(COCO)}$ | **11.27** |
| Ours + SD-unclip decoder | $0.08_{(COCO)}$ | **10.88** |

**Comparisons.** We compare our method with both the large-scale diffusion-based methods and the small-scale alignment methods. The results are provided in Table 1. By aligning text representations to image representations on the small MSCOCO data, our method achieves superior T2I generation than the large-scale methods,

Table 2: **Comparisons on various training data.** Our method consistently performs better.

| Method | CC3M | CC12M | LAION-HighRes 5M |
|---|---|---|---|
| Eclipse | 26.73 | 26.98 | 19.16 |
| Ours | **22.88** | **22.72** | **14.79** |

Karlo, Kandinsky, and Stable Diffusion without any diffusion steps. CS-Aligner also outperforms Eclipse and IB by an obvious margin using either Karlo or Kandinsky decoders. The results demonstrate the effective vision-language alignment capability of our method. Moreover, we compare CS-Aligner with Eclipse across different training datasets. As shown in Table 2, our method performs better across diverse training data (CC3M, CC12M, and LAION-HighRes-5M), underscoring the importance of the modality distribution information for robust alignment.

**Qualitative Visualization.** To further test our method, Fig. 4a shows qualitative visualizations of generated images using Karlo decoder. Our aligned text representations result in more realistic

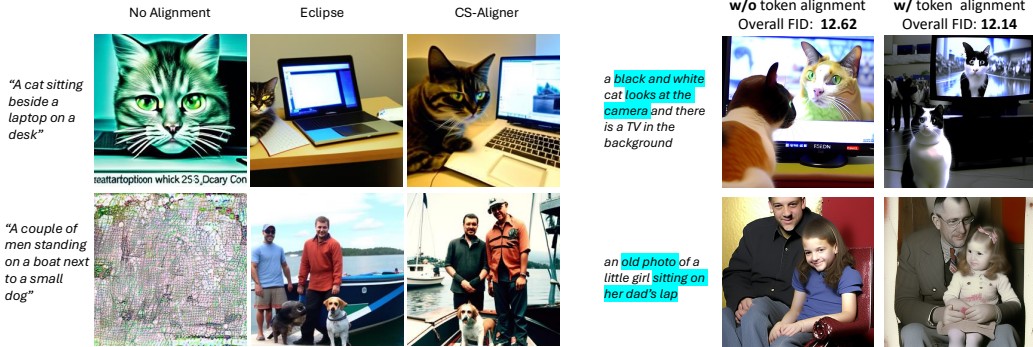

(a) **Qualitative comparison.** No alignment (left), Eclipse (middle), and CS-Aligner (right). CS-Aligner yields more realistic, semantically consistent generations.

(b) **CS-Aligner with token alignment.** Token alignment enhances fine-grained vision–language correspondence.

Figure 4: **Qualitative visualizations.**

images with stronger semantic consistency with the input sentence, highlighting the effectiveness of CS-Aligner in enhancing alignment. More visualizations are provided in Appendix F.1.

**CS-Aligner with different adaptation approaches.** To demonstrate the robustness of our method across different models, we perform alignments for T2I using both adapter and LoRA. Specifically, we apply LoRA with a low-rank dimension of 8 to every transformer layer in the CLIP text encoder. As shown in Table 3, based on different decoders, CS-Aligner with LoRA introduces fewer parameters, while still achieving comparable results compared with the

Table 3: **CS-Aligner with different adaptation approaches.** Our method achieves good alignment using both adapter and LoRA.

| Base Model | Adaptation | #Parameters | FID |
|---|---|---|---|
| Kandinsky | Adapter | 34M | 12.62 |
| | LoRA | 6M | 13.52 |
| Karlo | Adapter | 33M | 11.27 |
| | LoRA | 1.3M | 15.63 |

adapter-based one, showing the effectiveness and adaptability of CS-Aligner across different models.

**CS-Aligner with multiple captions.** It is common in real-world datasets for a single image to correspond to multiple captions (e.g., 5 captions per image in MSCOCO). Due to their pairwise alignment nature, previous methods such as InfoNCE and $\ell_2$-based approaches (Radford et al., 2021; Patel et al., 2024) struggle to simultaneously leverage multiple captions. In contrast, by incorporating CS divergence, our CS-Aligner enables training for alignment with single image and multiple captions through the divergence term. To demonstrate the benefits of multiple captions for CS-Aligner, we conducted experiments on the MSCOCO dataset by estimating the CS divergence term $\widehat{D}_{\text{CS}}$ in Eq. (6) using both single and multiple captions. As shown in Fig. 5a, CS-Aligner effectively leverages the information provided by multiple captions, leading to improved vision-language alignment.

**CS-Aligner with additional unpaired data.** Collecting and accurately annotating paired vision-language data is both challenging and costly. Enhancing alignment with additional unpaired data offers a more flexible and scalable solution for real-world applications. However, similar to the case of multiple captions, previous methods (Radford et al., 2021; Patel et al., 2024) struggle to fully utilize unpaired data due to their reliance on pairwise alignment, whereas CS-Aligner naturally incorporates the unpaired data information by CS divergence. To demonstrate this capability, we conduct experiments on the MSCOCO

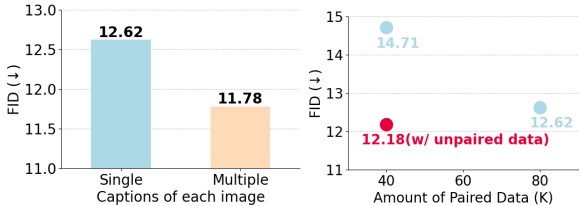

(a) Align with multi-captions. (b) Align with unpaired data.

Figure 5: **CS-Aligner with additional information.** Our method benefits from the additional information from multiple captions (a) and unpaired data (b).

dataset using the Kandinsky decoder with (1) 80K paired training samples, (2) 40K paired training samples, and (3) 40K paired training samples supplemented with 80K unpaired samples, where the unpaired samples are used to estimate the CS divergence. As shown in Fig. 5b, the result with

40K paired training data is lower than 80K. However, introducing additional unpaired data obviously improves the performance, even surpassing the model trained with 80K paired samples. This demonstrates CS-Aligner's ability to effectively leverage the distributional information of modalities for alignment.

**CS-Aligner with token alignment.** Beyond the unpaired data, CS-Aligner also enables token-level alignment by treating the tokens of each sample as a distribution. We evaluated the token-level extension of CS-Aligner with the Kandinsky decoder on MSCOCO. As shown in Fig. 4b, incorporating token alignment further improves performance. Moreover, qualitative results indicate that token alignment enhances fine-grained details in generated images, suggesting an improved ability to capture fine-grained relationships between modalities. Additional visualizations are provided in Fig. 7 in Appendix F.2.

## 4.2 IMAGE-TEXT RETRIEVAL

**Experimental Setup.** Effective multimodal alignment also benefits cross-modal retrieval.

To demonstrate the alignment ability of our method on retrieval tasks, we align LLMs (Dubey et al., 2024) text representations with CLIP vision representations on both image-to-text and text-to-image retrieval. We use the Flickr 1K test set (Young et al., 2014) for short-text retrieval, while Urban1K (Zhang et al., 2025) and DOCCI (Onoe et al., 2025) are employed for long-text retrieval. We compare CS-Aligner against pure

Table 4: **Comparisons of image-to-text (I2T) and text-to-image (T2I) retrieval.** Our method outperforms the baselines on diverse datasets.

| Methods | Flickr30k | | Urban-1k | | DOCCI | | Average | |
|---|---|---|---|---|---|---|---|---|
| | I2T | T2I | I2T | T2I | I2T | T2I | I2T | T2I |
| Long-CLIP | 90.0 | 76.2 | 82.5 | 86.1 | 66.5 | 78.6 | 79.7 | 80.3 |
| CLIP | 85.2 | 65.0 | 68.3 | 55.6 | 63.1 | 65.8 | 72.2 | 62.1 |
| LLM2CLIP-3M | 89.6 | 77.3 | 87.1 | 91.1 | 84.9 | 87.8 | 87.2 | 85.4 |
| Ours-3M | **91.8** | **81.0** | **87.6** | **92.2** | **86.6** | **89.1** | **88.7** | **87.4** |

InfoNCE-based methods, such as Long-CLIP (Zhang et al., 2025) and LLM2CLIP (Huang et al., 2024), as the baselines. To ensure a fair comparison, we adopt the setup from LLM2CLIP, aligning CLIP ViT-L/14 image representations with Llama 3 (8B) text embeddings. Both the vision and text representations are aligned by adapters trained on CC3M.

**Comparisons.** Table 4 shows that our method consistently and significantly outperforms the baselines across various datasets for both image-to-text (I2T) and text-to-image (T2I) retrieval. This demonstrates the effectiveness of our method for aligning two modalities into a shared space. Moreover, the ability to align a different text encoder (LLM) with the CLIP image encoder highlights the flexibility and generalizability of our approach.

## 5 RELATED WORK

**Vision-language alignment and applications.** CLIP (Radford et al., 2021) serves as a foundational model for vision-language alignment in multimodal tasks. Several works have enhanced CLIP through techniques such as momentum distillation (Li et al., 2021) and noisy text supervision (Jia et al., 2021). Despite its success, CLIP suffers from a persistent modality gap between text and image representations. Prior studies (Zhou et al., 2023; Liang et al., 2022; Shi et al., 2023) attribute this gap to factors such as cone effects (Liang et al., 2022) and suboptimal latent space structures (Shi et al., 2023). To address this, various strategies have been proposed, including projection adapters (Zhou et al., 2023; Gao et al., 2024; Huang et al., 2024), geodesic multimodal mixup (Oh et al., 2024), and parameter-efficient fine-tuning (Zanella & Ben Ayed, 2024). Recent works also improve CLIP by large language models (LLMs) (Jang et al., 2024; Koukounas et al., 2024; Huang et al., 2024) for downstream tasks such as **image-text retrieval**.

In addition to image-text retrieval, **text-to-image (T2I) generation** is another application that reflects the vision-language alignment capability. T2I has advanced significantly over the past decades, driven by both diffusion-based (Ramesh et al., 2021; Rombach et al., 2022; Saharia et al., 2022; Nichol et al., 2021) and GAN-based models (Zhang et al., 2017; Tao et al., 2023). Among diffusion-based methods, the unCLIP framework (Ramesh et al., 2021; 2022) employs a two-stage architecture with a CLIP-guided diffusion prior and a decoder (e.g., DALL-E-2 (Ramesh et al., 2022) or

Karlo (Donghoon et al., 2022)). Its prior module $g_\phi$ maps text representations $\mathbf{y}$ to image ones $\mathbf{x}$ by a diffusion model. Recently, Eclipse (Patel et al., 2024) employs an $\ell_2$ loss to simplify the prior loss by eliminating diffusion time and introducing a noise $\epsilon$ term: $\mathcal{L}_{\text{prior}} = \mathbb{E}_{\epsilon \sim \mathcal{N}(0,I)} \left[ \|\mathbf{x} - g_\phi(\epsilon, \mathbf{y})\|_2^2 \right]$. However, these methods still rely on pairwise loss (e.g., $\ell_2$). In contrast, our approach introduces distributional alignment for a more holistic modality alignment.

## 6  CONCLUSION

In this paper, we propose CS-Aligner, a novel distributional alignment framework that integrates Cauchy–Schwarz (CS) divergence with mutual information for multimodal alignment, which addresses the alignment and uniformity conflict of InfoNCE. By combining global distributional alignment with InfoNCE, CS-Aligner achieves tighter and more comprehensive alignment. By considering the modality distributional information, our method enables to leverage additional and detailed information from unpaired samples and tokens, leading to more flexible and fine-grained information for alignment. We demonstrate the effectiveness of our alignment on text-to-image generation and cross-modal retrieval.

### USE OF LARGE LANGUAGE MODELS (LLMS).

We used LLMs solely for minor language polishing. They were not involved in research ideation, experimental design, or substantive manuscript writing.

### ETHICS STATEMENT

Our proposed method advances research in multimodal alignment by introducing a novel distributional alignment approach. As a result, it also facilitates progress in multimodal generation. In the meantime, this capability may raise ethical concerns, including the potential misuse for generating deceptive or inappropriate content.

### REPRODUCIBILITY STATEMENT

We provide sufficient details for reproducibility in Sections 3 and G.

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

## A    DETAILS OF THE TOY EXAMPLES

***Example*** A.1.  Consider two Gaussian distributions, $p(\mathbf{x}) \sim \mathcal{N}(\mu_{\mathbf{x}}, \sigma_{\mathbf{x}}^2)$ and $p(\mathbf{y}) \sim \mathcal{N}(\mu_{\mathbf{y}}, \sigma_{\mathbf{y}}^2)$, with a joint distribution $p(\mathbf{x}, \mathbf{y}) \sim \mathcal{N}\left(\begin{pmatrix} \mu_{\mathbf{x}} \\ \mu_{\mathbf{y}} \end{pmatrix}, \begin{pmatrix} \sigma_{\mathbf{x}}^2 & \rho\sigma_{\mathbf{x}}\sigma_{\mathbf{y}} \\ \rho\sigma_{\mathbf{x}}\sigma_{\mathbf{y}} & \sigma_{\mathbf{y}}^2 \end{pmatrix}\right)$. Here, $\mu_{\mathbf{x}}$ and $\mu_{\mathbf{y}}$ are the means of $\mathbf{x}$ and $\mathbf{y}$, $\sigma_{\mathbf{x}}^2$ and $\sigma_{\mathbf{y}}^2$ are their variances, and $\rho$ is the correlation coefficient and controls their linear dependency. When $\rho = 0.99$, the two modalities are highly dependent, with high mutual information ($I = 1.959$; see Fig. 2a and 2b). When $\rho = 0$, the modalities are independent, resulting in zero mutual information (Fig. 2c). Interestingly, two distributions with the same mutual information value can either exhibit minimal statistical distance and nearly identical shapes, including similar locations, widths, and higher-order moments, as shown in Fig. 2b, or have completely different shapes with distinct means (0 for $p(\mathbf{x})$ and 2 for $p(\mathbf{y})$) and variances (4 for $p(\mathbf{x})$ and 1 for $p(\mathbf{y})$), as illustrated in Fig. 2a. Quantitatively, the former case shows a minimal KL divergence of 0, while the latter exhibits a KL divergence of nearly $6.81$.

**Mutual information.**   For two continuous random variables $\mathbf{x}$ and $\mathbf{y}$, the mutual information is defined as:

$$I(\mathbf{x}; \mathbf{y}) = \iint p(\mathbf{x}, \mathbf{y}) \log\left(\frac{p(\mathbf{x},\mathbf{y})}{p(\mathbf{x})\,p(\mathbf{y})}\right) d\mathbf{x}\, d\mathbf{y}. \tag{13}$$

For a bivariate Gaussian distribution

$$p(\mathbf{x}, \mathbf{y}) \sim \mathcal{N}\left(\begin{pmatrix} \mu_{\mathbf{x}} \\ \mu_{\mathbf{y}} \end{pmatrix}, \begin{pmatrix} \sigma_{\mathbf{x}}^2 & \rho\sigma_{\mathbf{x}}\sigma_{\mathbf{y}} \\ \rho\sigma_{\mathbf{x}}\sigma_{\mathbf{y}} & \sigma_{\mathbf{y}}^2 \end{pmatrix}\right),$$

the mutual information admits the closed-form solution:

$$I(\mathbf{x}; \mathbf{y}) = -\tfrac{1}{2} \ln\left(1 - \rho^2\right). \tag{14}$$

In particular, for correlation $\rho = 0.99$, we have $I(\mathbf{x}, \mathbf{y}) \approx 1.959$, while for $\rho = 0$, the variables are independent and $I(\mathbf{x}, \mathbf{y}) = 0$.

**Divergence.**   For univariate Gaussian distributions $p(\mathbf{x}) = \mathcal{N}\left(\mu_{\mathbf{x}}, \sigma_{\mathbf{x}}^2\right)$ and $p(\mathbf{y}) = \mathcal{N}\left(\mu_{\mathbf{y}}, \sigma_{\mathbf{y}}^2\right)$, the KL divergence is given by:

$$D_{\mathrm{KL}}\left(p(\mathbf{x}) \,\|\, p(\mathbf{y})\right) = \ln\left(\frac{\sigma_{\mathbf{y}}}{\sigma_{\mathbf{x}}}\right) + \frac{\sigma_{\mathbf{x}}^2 + (\mu_{\mathbf{x}} - \mu_{\mathbf{y}})^2}{2\,\sigma_{\mathbf{y}}^2} - \tfrac{1}{2}. \tag{15}$$

For Fig. 2b and Fig. 2c, we set $\sigma_{\mathbf{x}} = \sigma_{\mathbf{y}} = 1$. Hence, when $\mu_{\mathbf{x}} = \mu_{\mathbf{y}} = 0$, $D_{\mathrm{KL}}\left(p(\mathbf{x}) \,\|\, p(\mathbf{y})\right) = 0$.

For Fig. 2a, we use $\sigma_x = 2$ and $\sigma_y = 1$. When $\mu_{\mathbf{x}} = 0$ and $\mu_{\mathbf{y}} = 2$, the $D_{\mathrm{KL}}\left(p(\mathbf{x}) \,\|\, p(\mathbf{y})\right) \approx 6.81$, which is very large.

## B    DERIVATIONS

In this section, we provide a derivation of alignment and uniformity terms of InfoNCE. More concrete analysis can be found in (Wang & Isola, 2020).

Let $(\mathbf{x}, \mathbf{y})$ be positive (image–text) pairs drawn from $p_{\mathrm{pair}}$, and let $\{(\mathbf{x}_i', \mathbf{y}_i')\}_{i=1}^M$ be $M$ negative samples (unpaired samples) drawn i.i.d. from the marginal $p_{\mathrm{data}}$. The one-sided InfoNCE (CLIP) loss with temperature $\tau > 0$ is

$$\mathcal{L}_{\mathrm{InfoNCE}} = -\frac{1}{2}\,\mathbb{E}_{(\mathbf{x},\mathbf{y}) \sim p_{\mathrm{pair}}} \mathbb{E}_{\{\mathbf{x}_i', \mathbf{y}_i'\} \sim p_{\mathrm{data}}} \left[\log \frac{e^{\mathbf{x}^\top \mathbf{y}/\tau}}{\sum_{i=1}^M e^{\mathbf{x}_i'^\top \mathbf{y}/\tau}} + \log \frac{e^{\mathbf{x}^\top \mathbf{y}/\tau}}{\sum_{i=1}^M e^{\mathbf{x}^\top \mathbf{y}_i'/\tau}}\right].$$

In CLIP, the features are normalized to compute the loss. Under this unit-norm constraint $\|\mathbf{x}\|_2 = \|\mathbf{y}\|_2 = 1$, $\mathcal{L}_{\mathrm{InfoNCE}}$ decomposes into

$$\mathcal{L}_{\mathrm{InfoNCE}} = \underbrace{-\mathbb{E}_{(\mathbf{x},\mathbf{y}) \sim p_{\mathrm{pair}}}\left[\frac{\mathbf{x}^\top \mathbf{y}}{\tau}\right]}_{\mathcal{L}_{\mathrm{align}}} + \underbrace{\mathbb{E}_{(\mathbf{x},\mathbf{y}) \sim p_{\mathrm{data}}}\left[\tfrac{1}{2}\log \sum_{i=1}^M e^{\mathbf{x}^\top \mathbf{y}_i'/\tau} + \tfrac{1}{2}\log \sum_{i=1}^M e^{\mathbf{x}_i'^\top \mathbf{y}/\tau}\right]}_{\mathcal{L}_{\mathrm{uniform}}},$$

up to an additive constant. Moreover, by writing

$$\|\mathbf{x} - \mathbf{y}\|_2^2 = \|\mathbf{x}\|^2 + \|y\|^2 - 2\,\mathbf{x}^\top \mathbf{y} = 2 - 2\,\mathbf{x}^\top \mathbf{y} \quad \Longrightarrow \quad \mathbf{x}^\top \mathbf{y} = 1 - \tfrac{1}{2}\|\mathbf{x} - \mathbf{y}\|_2^2, \tag{16}$$

we can show that:

**(i)** *Alignment.*

$$-\mathbb{E}_{(\mathbf{x},\mathbf{y})}\Big[\frac{\mathbf{x}^\top \mathbf{y}}{\tau}\Big] = -\mathbb{E}\Big[\frac{1 - \tfrac{1}{2}\|\mathbf{x} - \mathbf{y}\|^2}{\tau}\Big] = -\frac{1}{\tau} + \frac{1}{2\tau}\,\mathbb{E}\big[\|\mathbf{x} - \mathbf{y}\|^2\big].$$

Dropping the constant $-1/\tau$, define

$$\mathcal{L}_{\text{align}} := \frac{1}{2\tau}\,\mathbb{E}_{(\mathbf{x},\mathbf{y})\sim p_{\text{pair}}}\big[\|\mathbf{x} - \mathbf{y}\|_2^2\big]. \tag{17}$$

**(ii)** *Uniformity.* For each negative (unpaired) sample $y_i'$, using Eq. equation 16,

$$e^{\mathbf{x}^\top \mathbf{y}_i'/\tau} = e^{(1 - \tfrac{1}{2}\|\mathbf{x} - \mathbf{y}_i'\|^2)/\tau} = e^{1/\tau}\, e^{-\frac{1}{2\tau}\|\mathbf{x} - \mathbf{y}_i'\|^2}.$$

Hence

$$\sum_{i=1}^{M} e^{\mathbf{x}^\top \mathbf{y}_i'/\tau} = e^{1/\tau} \sum_{i=1}^{M} e^{-\frac{1}{2\tau}\|\mathbf{x} - \mathbf{y}_i'\|^2}, \quad \log \sum_i e^{\mathbf{x}^\top \mathbf{y}_i'/\tau} = \tfrac{1}{\tau} + \log \sum_i e^{-\frac{1}{2\tau}\|\mathbf{x} - \mathbf{y}_i'\|^2}.$$

An identical argument holds for the $\{\mathbf{x}_i', \mathbf{y}\}$ terms. Up to constants,

$$\mathcal{L}_{\text{uniform}} := \mathbb{E}_{(\mathbf{x},\mathbf{y})\sim p_{\text{data}}}\Big[\tfrac{1}{2}\log \sum_{i=1}^{M} e^{-\frac{1}{2\tau}\|\mathbf{x} - \mathbf{y}_i'\|^2} + \tfrac{1}{2}\log \sum_{i=1}^{M} e^{-\frac{1}{2\tau}\|\mathbf{x}_i' - \mathbf{y}\|^2}\Big]. \tag{18}$$

In the limit of large batch size one may further rewrite

$$\mathcal{L}_{\text{uniform}} \approx \log \mathbb{E}_{\mathbf{x},\mathbf{y}\sim p_{\text{data}}}\big[\exp(-t\|\mathbf{x} - \mathbf{y}\|_2^2)\big],$$

with $t = \frac{1}{2\tau}$.

Combining (i) and (ii) and absorbing all additive constants gives the desired decomposition

$$\boxed{\mathcal{L}_{\text{clip}} = \mathcal{L}_{\text{align}} + \mathcal{L}_{\text{uniform}} + \text{const.}}$$

## C  RELATED WORK OF CAUCHY-SCHWARZ (CS) DIVERGENCE.

CS divergence (Principe et al., 2000a;b) is derived from the Cauchy-Schwarz inequality for square-integrable functions. It serves as a symmetric distribution distance metric with notable properties, such as the ability to measure conditional distributions (Yu et al., 2025) and the closed-form expression for mixtures of Gaussians (Kampa et al., 2011). CS divergence has been successfully applied across various domains, including deep clustering (Trosten et al., 2021), disentangled representation learning (Tran et al., 2022), and deep regression (Yu et al., 2024). Moreover, due to its advantage of estimating discrepancy between conditional distributions, it has demonstrated success in the domain adaption area (Yin et al., 2024) and time series clustering (Yu et al., 2025). However, the utility of CS divergence in foundation models remains unclear and unexplored.

## D  COMPARISON BETWEEN CS DIVERGENCE AND OTHER METRICS

Unlike parametric distributions, distributions of different real-world modalities exhibit unpredictable variability and inconsistent overlaps, meaning that $p(\mathbf{x})$ and $p(\mathbf{y})$ may follow arbitrary distributions with a small intersection. Therefore, it is crucial to overcome these challenges to measure and optimize multimodal distribution divergence robustly. Below, we outline several key properties that an effective metric should satisfy for multimodal alignment.

***Remark*** D.1.  Key properties for distribution align metrics:

- *Symmetry*: Both distributions are treated equally, ensuring consistent and unbiased multi-modal alignment, formulated by $D(p(\mathbf{x}), p(\mathbf{y})) = D(p(\mathbf{y}), p(\mathbf{x}))$.

- *Differentiable and Efficient Estimation*: Enable differentiable estimation without distribution assumptions to facilitate optimization, formulated as $\partial D(p(\mathbf{x}; \theta), p(\mathbf{y}; \phi)) \neq \emptyset, \forall p(\mathbf{x}), p(\mathbf{y})$. Achieve the estimation non-parametrically or efficiently.

- *Robustness to Small Distribution Overlap*: Provide reliable measurements even when distributions have minimal overlap of supports, which may often occur in multimodal scenarios. The property is formulated as $0 \leq D(p(\mathbf{x}), p(\mathbf{y})) \leq \infty$ when $0 < \mu\big(\text{supp}(p(\mathbf{x})) \cap \text{supp}(p(\mathbf{y}))\big) < \epsilon$. $\mu\big(\text{supp}(p(\mathbf{x})) \cap \text{supp}(p(\mathbf{y}))\big)$ denotes the overlap of $p(\mathbf{x})$ and $p(\mathbf{y})$. $\epsilon$ is a small value.

These properties enable the divergence term to align arbitrary distributions with small support overlap, which is well-suited for large-scale multimodal applications involving deep learning.

### D.1 CONNECTION TO THE PRIOR LOSS

**Remark** D.2. Connection to the prior loss ($\ell_2$ loss) used by Eclipse (Patel et al., 2024):

$$\mathcal{L}_{\text{prior}} = \mathbb{E}_{\epsilon \sim \mathcal{N}(0,I)} \left[ \|\mathbf{x} - g_\phi(\epsilon, \mathbf{y})\|_2^2 \right]. \tag{19}$$

Consider the third term in Eq. (8), which involves $\kappa(\mathbf{x}_i, \mathbf{y}_j)$ defined by the Gaussian kernel $\kappa_\sigma(\mathbf{x}, \mathbf{y}) = \exp\big(-\|\mathbf{x} - \mathbf{y}\|_2^2/2\sigma^2\big)$. A second-order Taylor expansion yields

$$\kappa(\mathbf{x}_i, \mathbf{y}_j) = \exp\left( -\frac{(\mathbf{x}_i - \mathbf{y}_j)^2}{2\sigma^2} \right) \approx 1 - \frac{(\mathbf{x}_i - \mathbf{y}_j)^2}{2\sigma^2}. \tag{20}$$

When $i = j$ (i.e., diagonal of $\kappa(\mathbf{x}, \mathbf{y})$), this approximation reduces to a weighted $\ell_2$ loss by $1/2\sigma^2$, analogous to the Eq. 19. Consequently, the $\ell_2$ loss emerges as a special case of our divergence, focusing solely on paired sample reconstruction and omitting broader distribution alignment, including off-diagonal (cross-sample) contributions.

### D.2 COMPARISON WITH KL DIVERGENCE.

KL divergence is a widely used metric in deep learning. Given two distributions, $p(\omega)$ and $q(\omega)$, the KL divergence is defined as:

$$D_{\text{KL}}(p; q) = \int p(\omega) \log \frac{p(\omega)}{q(\omega)} \, d\omega. \tag{21}$$

**Validity for multimodal alignment.** Define the support sets of distributions $p$ and $q$ as:

$$\text{supp}(p) = \{\omega \in \Omega : p(\omega) > 0\}, \quad \text{supp}(q) = \{\omega \in \Omega : q(\omega) > 0\}. \tag{22}$$

For KL divergence, if there exists any point $x \in \text{supp}(p)$ such that $q(x) = 0$, the term $p(\omega) \log \frac{p(\omega)}{0} \to \infty$, leading to: $D_{\text{KL}}(p; q) = \infty$. Thus, a necessary condition for KL divergence to be finite is $\text{supp}(p) \subseteq \text{supp}(q)$. Otherwise, KL divergence becomes invalid.

In contrast, the CS divergence becomes infinite only if there is no overlap between supports of $p$ and $q$, i.e., when $\int p(\omega)q(\omega)d\omega = 0$, making the logarithm undefined. Hence, the condition for finite CS divergence is: $\text{supp}(p) \cap \text{supp}(q) \neq \emptyset$.

In multimodal alignment, it's reasonable to assume that the two modality distributions partially overlap but are not disjoint, as supported by our empirical observations in Fig. 4a (no alignment results). Under these conditions, KL divergence can be invalid and therefore suboptimal. Conversely, the CS divergence condition is less restrictive, making it more suitable and stable for multimodal alignment.

**Compatibility with InfoNCE** Integrating InfoNCE with CS divergence explicitly encourages intra-modality uniformity and cross-modality alignment, thereby effectively improving multimodal alignment. For KL divergence, assuming the distributions of the two modalities are Gaussian, $\mathcal{N}(\mu_0, \Sigma_0)$

and $\mathcal{N}(\mu_1, \Sigma_1)$, the divergence can be computed as:

$$\mathcal{D}_{\text{KL}}\left[\mathcal{N}(\mu_0, \Sigma_0) \| \mathcal{N}(\mu_1, \Sigma_1)\right] = \frac{1}{2}\left(\text{tr}\left(\Sigma_1^{-1}\Sigma_0\right) + (\mu_1 - \mu_0)^\top \Sigma_1^{-1}(\mu_1 - \mu_0) - k + \log\left(\frac{\det \Sigma_1}{\det \Sigma_0}\right)\right).$$
(23)

This formulation lacks explicit connections to the InfoNCE in terms of alignment and uniformity, making it less compatible with InfoNCE compared to the CS divergence.

**Nonparametric estimation.** Additionally, when the distributions are not assumed to be Gaussian, a nonparametric estimator is required for KL divergence. A common choice, the k-NN estimator (Wang et al., 2009), is non-differentiable, which poses challenges for optimization in gradient-based learning frameworks. In contrast, the CS divergence demonstrates greater stability and differentiability when paired with KDE, making it a more robust choice.

**Experimental Comparison.** To verify the above analysis, we compare CS divergence and KL divergence on the unpaired data scenario, where KL can easily become invalid. We trained a KL + InfoNCE model in our unpaired data setting—using paired data for InfoNCE and unpaired data for divergence. The initial KL value exceeded 5000 (extremely large), and consequently, the model could not converge, leading to catastrophic failure. In contrast, CS divergence remained stable (initial value around 3), and achieved comparable final performance with an FID of 12.18 (Fig. 5b in the main paper).

## D.3 COMPARISON WITH WASSERSTEIN DISTANCE.

Wasserstein Distance is also widely used for distribution discrepancy (e.g. GAN (Arjovsky et al., 2017)). However, Wasserstein distance is be computed either by using an additional learnable module (e.g., a neural network for estimating a transport map (Korotin et al., 2022)) or by solving an optimization problem, often approximated via multiple Sinkhorn (Cuturi, 2013) iterations for computational efficiency, leading to efficiency problem in large-scale training. In contrast, CS divergence can be efficiently estimated by a nonparametric estimator.

## D.4 QUANTITATIVE COMPARISONS WITH KL AND WASSERSTEIN DISTANCE.

We compare our method with KL and Wasserstein distances below. To make the KL divergence tractable, we assume the batch embeddings follow Gaussian distributions. For the Wasserstein distance, we either use the closed-form Gaussian Wasserstein distance under the same assumption or apply the Sinkhorn algorithm for general distributions. However, in practice, we found that Sinkhorn often fails to converge. The results show that our method outperforms both KL and Wasserstein distances. Moreover, Wasserstein distance and KL lack an InfoNCE-style alignment–uniformity decomposition; only CS-divergence yields the compatible formulation (Eq. 10). The Gaussian assumption is also stronger than our nonparametric method.

| Method | FID $\downarrow$ |
|---|---|
| KL | 23.48 |
| W-distance | 18.41 |
| Sinkhorn | Not converge |
| CS-Aligner | 12.62 |

## D.5 COMPARISON WITH MUTUAL INFORMATION DIVERGENCE (KIM ET AL., 2022).

Mutual information estimation depends on parametric assumptions about the underlying distributions, e.g., multivariate Gaussian, whereas CS divergence imposes no such constraints. Moreover, estimating mutual information decomposes into a mutual information term plus two KL divergences, and thus lacks explicit connections to the InfoNCE in terms of alignment and uniformity.

## E   COMPUTATION COMPLEXITY AND STABILITY ANALYSIS

We normalize high-dimensional embeddings onto the unit hypersphere and use a fixed Gaussian kernel bandwidth so that concentration of measure and classical KDE theory ensure stable, low-variance estimates.

In high dimensions, mapping embeddings onto the unit hypersphere exploits the concentration of measure phenomenon: as $d$ grows, the pairwise distances $\|x - y\|$ between random points on $S^{d-1}$ concentrate sharply around $\sqrt{2}$, with fluctuations of order $O(1/\sqrt{d})$. Consequently, a Gaussian kernel

$$K(x,y) = \exp\left(-\frac{\|x - y\|^2}{2\sigma^2}\right), \tag{24}$$

with fixed bandwidth (e.g. $\sigma = 1$) yields values confined to a narrow, well-behaved range, preventing weights from collapsing to 0 or saturating at 1 and ensuring smoothly varying density estimates (Berestycki & Nickl, 2009).

Moreover, when the effective sample size $n$ (e.g. batch size) and dimensionality $d$ satisfy $n\,\sigma^d \gg 1$, which holds for $\sigma = 1$, $n \sim 10^3$, and $d \sim 10^3$, the KDE estimator obeys a central limit theorem. This guarantees that CS divergence estimates have vanishing variance and stable gradients during optimization (Parzen, 1962).

**Computational complexity.**   The computation cost of our method is comparable to the CLIP-based method when scaling up to even larger-scale datasets. The computation complexity is $O(N^2)$, which is the same as the InfoNCE used in CLIP. However, the computational complexity is feasible to scale up to larger-scale datasets.

## F   MORE RESULTS

### F.1   MORE VISUALIZATION

We illustrate more high-resolution images generated by the Kandinsky decoder with our aligned text representation in Fig. 6. The adapter is trained on LAION-HighRes 5M.

### F.2   MORE VISUALIZATION FOR TOKEN ALIGNMENT

We provide more visualizations with and without the token alignment Fig. 7, demonstrating its ability to generate more fine-grained images with CS-Aligner.

## G   IMPLEMENTATION DETAILS

**Implementation details**   Our models were trained on 4 NVIDIA RTX A100 GPUs with a global batch size of 1,024 (256 per GPU). We optimized parameters using AdamW with a cosine annealing learning rate schedule, spanning a total of 100 GPU hours. Mixed-precision training (FP16) was employed to enhance computational efficiency while maintaining stability. We use the learning rate of $5e - 5$. We use hyperparameter $\lambda$ as 0.01 to keep the same number scale as the divergence.

**Kernel density estimator.**   A proper kernel size is critical in KDE for accurate estimation of Eq. (8). In this paper, we follow Yin et al. (2024) to normalize the features from two modalities and use a kernel size 1. In general, this is sufficient to ensure stable learning.

### G.1   T2I DETAILS

**Figure 1 implementation details.**   For Fig. 1a and Fig. 1b, we train the same adapter on top of the CLIP model using InfoNCE and CS-Aligner, respectively. We use the MSCOCO training set and visualize the learned representations with t-SNE on 5K image–text pairs from the validation set. For the temperature in both InfoNCE and CS-Aligner, we initialize it from the pretrained CLIP model and keep it learnable during training. For Fig. 1c, we compute the L2 distance between

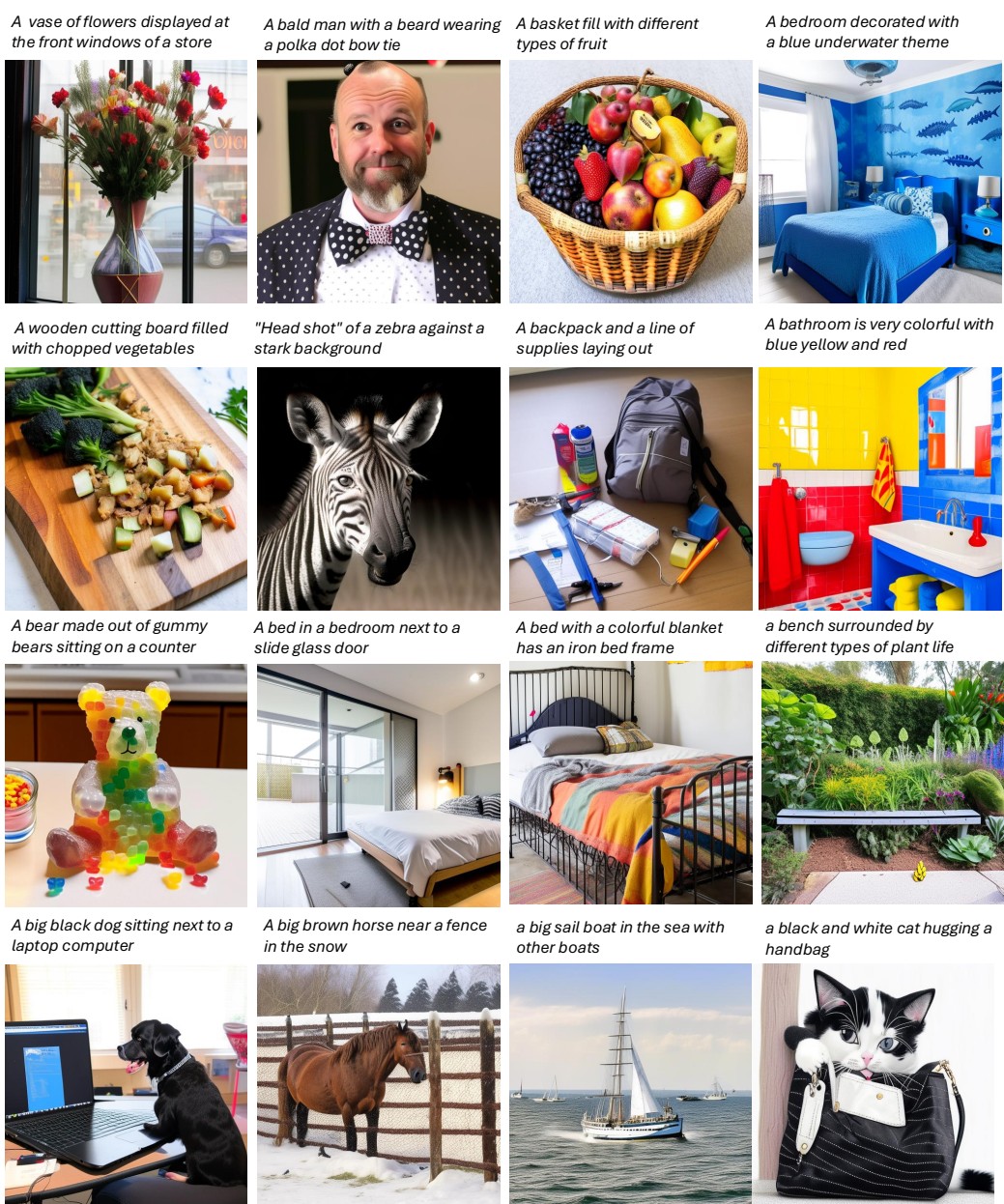

Figure 6: **Qualitative visualization.** The adapter is trained on LAION-HighRes 5M. The aligned text representation is then decoded by the Kandinsky decoder.

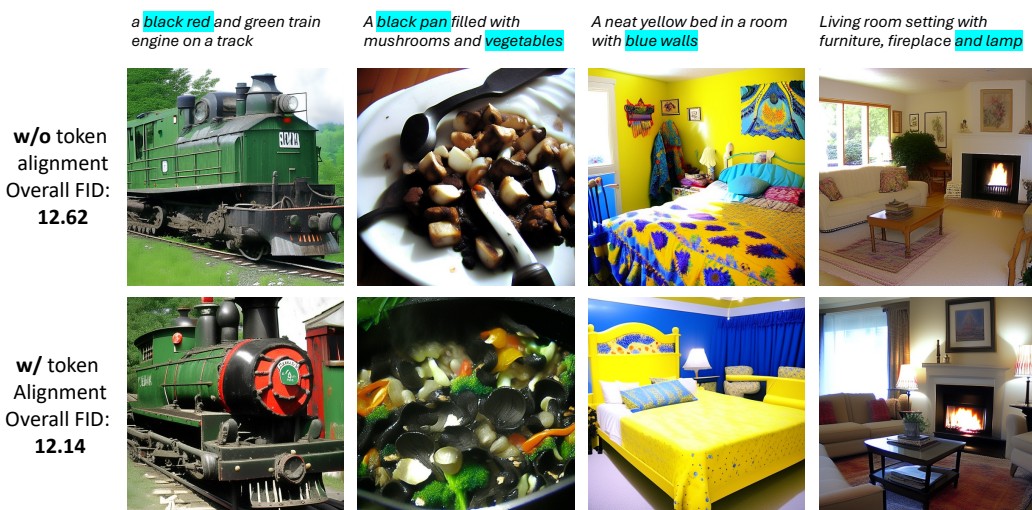

Figure 7: **Token alignment is effective for fine-grained generations with more details and stronger semantic correspondence with the text inputs.**

the embeddings of all image–text pairs and visualize the resulting histogram. The histogram of L2 distances for positive pairs systematically reflects their distance distribution.

**Kandinsky details.** We use Kandinsky v2.2, an unCLIP-type model that utilizes CLIP ViT-bigG-14-laion2B-39B-b160k with 1280 projection dimensions for text and image encoders. Kandinsky v2.2 employs a latent diffusion model and MOVQ (Zheng et al., 2022) as the decoder to generate images of size $512 \times 512$ from the given image representation. When using the Kandinsky decoder, we apply 50 denoising steps (Ho et al., 2020) with a classifier-free guidance scale of 7.5 (Ho & Salimans, 2022).

**Karlo details.** Karlo uses CLIP-ViT-L/14 with 768 projection dimensions for image and text encoders. It employs a diffusion model to decode the image representation into a low-resolution image, followed by a super-resolution diffusion module that upsamples it to $512 \times 512$. When using the Karlo decoder, we apply 25 denoising steps with a classifier-free guidance scale of 7.5, followed by an additional 7 super-resolution steps.

**Adapter details.** To ensure a fair comparison, our adapter module has the same architecture as Eclipse (Patel et al., 2024), which is based on the standard PriorTransformer model (Ramesh et al., 2022) but modified to be time-independent. Specifically, it consists of 10 layers with 16 attention heads, each having a head dimension of 32. The embedding dimension is 768/1280, with three additional embeddings. The model does not use time embeddings and has a dropout rate of 0.0. For the text to image generation task, in order to use the pretrained image generator, we only use the text adapter. For the retrieval and classification, we use adapters for both modalities.

**LoRA** We configure LoRA (Low-Rank Adaptation) for CLIP with a rank of $r = 8$ and a scaling factor of $\alpha = 16$, enabling efficient adaptation while maintaining a low computational footprint. The targeted modules include the self-attention projections, the fully connected layers, and the `text_projection` layer, ensuring adaptation across both vision and text processing components. A dropout rate of $0.1$ is applied to enhance regularization. For the CLIP encoder in Kandinsky, ViT-bigG-14-laion2B-39B-b160k, the number of LoRA parameters is 6 million. As for CLIP-ViT-L/14 in Karlo, the CLIP model size is smaller, resulting in $1.3$ million LoRA parameters.

**LAION-HighResolution-5M selection.** We use a subset of 5 million image-text pairs from the LAION-HighResolution dataset, which contains 175 million pairs. Due to computational constraints, we download only a portion of the dataset and select pairs with English captions.

# H    MORE EXPERIMENTAL RESULTS

**Image-text classification.**    We compare with CLIP-Adapter (Gao et al., 2024) on the image classification task following their few-shot classification setting. We fine-tune the adapter based on ViT-B/16 with 16-shots subset for each of the 11 datasets. The results are provided in the following table. With better alignment, our method consistently performs better.

Table 5: **Comparison with CLIP-Adapter on the image classification task.** Our methods performs consistently better on various datasets.

| Method | ImageNet | Caltech101 | DTD | EuroSAT | FGVCAircraft | Food101 | Flowers102 | OxfordPets | StanfordCars | SUN397 | UCF101 | Average |
|---|---|---|---|---|---|---|---|---|---|---|---|---|
| (Gao et al., 2024) | 71.1 | 94.4 | 70.9 | 85.7 | 42.8 | 83.2 | 96.0 | 92.1 | 78.6 | 75.0 | 82.8 | 79.3 |
| Ours | **72.9** | **95.0** | **72.3** | **87.2** | **44.4** | **85.8** | **97.5** | **93.0** | **81.9** | **76.2** | **84.0** | **80.9** |

Table 6: **Image captioning results.**

| Method | Bleu_1 $\uparrow$ | CIDEr $\uparrow$ |
|---|---|---|
| InfoNCE+LM | 40.4 | 14.3 |
| InfoNCE+LM+CS | 41.3 | 16.7 |

**Image captioning.**    We extend our method to the image captioning task. We adopt the Blip2 (Li et al., 2023a) stage one training strategy to highlight the importance of representation alignment for the image captioning task. We train a Q-former with the image text contrastive loss (InfoNCE) and the language model loss on the MSCOCO captioning dataset. The results in Table 6 show that our method can improve the image captioning ability. Also, the qualitative results of image captioning are shown in Fig. reffig:vis-captioning. The generated captions are semantically aligned with the images, demonstrating the general applicability of our method.

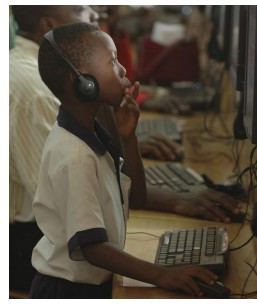 Captioning: A young boy Wearing headphones.

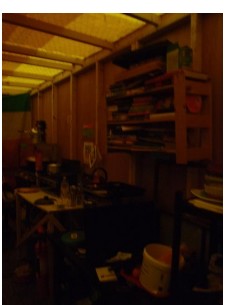 Captioning: A cluttered room with a table and chairs in it.

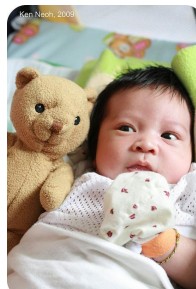 Captioning: A baby laying in a bed with a teddy bear.

Figure 8: Qualitative results of image captioning.

## H.1    ABLATION STUDY

**Hyperparameter Sensitivity Analysis.**    We conducted a sensitivity analysis on the two key hyperparameters, $\lambda$ (the weight for InfoNCE) and $\sigma$ (the Gaussian kernel width). For efficiency, we evaluated on a subset of 10 000 MSCOCO training samples and report Fréchet Inception Distance (FID) as the metric.

Table 7 shows that our method is robust to moderate variations in both $\lambda$ and $\sigma$, with only minor FID fluctuations over a wide range. A large $\lambda$ overemphasizes distributional alignment, optimizing intra-modality uniformity and global distribution distance while overlooking the pairwise alignment term. Since the generation task is sensitive to both global distribution closeness and sample-wise alignment, an excessively large $\lambda$ can degrade performance. We also evaluate the sensitivity of $\lambda$ and $\sigma$ on the MSCOCO retrieval task. The results show that our method is robust and performs well across a wide range of hyperparameters.

Table 7: **Sensitivity of FID to $\lambda$ and $\sigma$.**

| $\lambda$ | 0.01 | 0.1 | 1 | 10 |
|-----------|------|-----|---|----|
| FID | 81.34 | 32.51 | 29.86 | 65.79 |

(a) $\lambda$ sensitivity

| $\sigma$ | 0.1 | 0.5 | 1 | 1.5 |
|----------|-----|-----|---|-----|
| FID | 30.58 | 27.79 | 29.86 | 31.49 |

(b) $\sigma$ sensitivity

Table 8: **Sensitivity to $\lambda$ and $\sigma$. on MSCOCO retrieval**

| $\sigma$ | 0.1 | 0.5 | 1 | 1.5 |
|----------|-----|-----|---|-----|
| R@1 | 49.3 | 50.9 | 50.7 | 50.2 |

| $\lambda$ | 0.01 | 0.1 | 1 | 10 |
|-----------|------|-----|---|----|
| R@1 | 50.1 | 50.6 | 50.7 | 48.6 |

**Alignment with InfoNCE is not enough for the generation task.** We ablate InfoNCE and In-foNCE with CS divergence (CS-Aligner) on the text-to-image generation task. Specifically, we train the adapter on the MSCOCO dataset and use the Kandinsky decoder to generate the corresponding images. For the InfoNCE temperature, we resume it from the pretrained CLIP model and keep it learnable. We then compute the FID score for comparison (lower is better). Table 9 shows that InfoNCE alone struggles to align the multimodal distributions, resulting in a high FID score. As the learnable temperature $\tau$ (inherited from CLIP) decreases during training, the contrastive logits become sharper, making the uniformity term dominate over the alignment term and thereby weakening multimodal alignment (see our decomposition in Sec. 2) For text-to-image generation, the decoder requires the two modalities to lie in the same distribution, which InfoNCE alone is unable to guarantee. Hence, an InfoNCE-only model may still perform well in cosine-similarity–based retrieval but fails in generation due to the persistent distributional gap.

We also provide the retrieval ablations. Retrieval requires only correct relative similarity ranking, not full distributional overlap, so the degradation of InfoNCE-only is smaller. Nevertheless, CS-Aligner consistently outperforms InfoNCE-only, likely because the intra-modality uniformity terms (Eq. 9) promote better sample separability, which benefits retrieval.

Table 9: **Ablation study of CS-Aligner on retrieval and generation.** Alignment with CS-Aligner significantly outperforms using InfoNCE alone.

| Method | Retrieval | | | Generation |
|--------|-----------|-----|-----|------------|
| | T2I | I2T | Avg | FID $\downarrow$ |
| InfoNCE | 50.1 | 65.8 | 57.95 | 151.35 |
| CS-Aligner | 50.7 | 66.34 | 58.52 | 12.62 |

