# OpenReview forum: "Distributional Vision-Language Alignment by Cauchy-Schwarz Divergence"
_ICLR.cc/2026/Conference — ICLR 2026 Poster_

### Official Review · Reviewer_qrYc · 2025-10-22

**Soundness:** 3
**Presentation:** 3
**Contribution:** 3
**Rating:** 6
**Confidence:** 4

**Summary:**

This paper argues that the widely used InfoNCE in CLIP learning ignores distributional differences and has inherent conflict in terms of alignment and uniformity, which result in suboptimal alignment and modality gap. To tackle these limitations, the paper proposes CS-Aligner, that integrates Cauchy-Schwarz divergence with InfoNCE, leverages unpaired data and incorporates distributional token-wise alignment. Experiments on text-to-image generation and cross-modality retrieval tasks demonstrate the effectiveness of CS-Aligner

**Strengths:**

1. The paper proposes a novel framework, CS-Aligner, to reduce modality gap for better vision-language alignment.
2. The idea of using Cauchy-Schwarz divergence is ingenious, which can simultaneously tackle the two limitations: ignoring of distributional differences and alignment/uniformity confliction. Authors also provide comprehensive theoretical analysis to support it.
3. Experiments on several multi-modal/cross-modal tasks show the effectiveness of CS-Aligner.

**Weaknesses:**

1. The Cauchy-Schwarz divergence constitutes the core contribution of this work. It is therefore essential to thoroughly compare it against other distribution distance measures—such as KL/JS divergence and the Wasserstein distance. While the authors discuss their differences in the Appendix, they provide no quantitative comparisons.

2. A main ablation table should be included to quantitatively evaluate the effectiveness of each component in CS-Aligner. Moreover, the ablation study should extend beyond text-to-image generation to include more tasks, which would help comprehensively assess the impact of individual components.

3. The authors should clarify how the visualizations in Figure 1 were derived, particularly regarding the dataset and experimental settings used. For qualitative analysis, it would be beneficial to also present results on other tasks—such as cross-modal retrieval and image captioning—to better demonstrate the method's general applicability.

4. Regarding Table 7, why larger lambda leads to worse results? How about the sensitivity of lambda and sigma on other tasks?

Minor issue: In Line 431, cross-model -> cross-modal

**Questions:**

See Weaknesses.

---

> ### Author Response · Authors · 2025-11-20
>
> *We sincerely thank #Reviewer qrYc for their insightful comments. The following addresses their concerns and provides answers to their questions.*
>
> ---
>
> **1. Quantitative comparisons with KL and Wasserstein distance.**
>
> We compare our method with the KL and Wasserstein distances below. To make the KL divergence tractable, we assume the batch embeddings follow Gaussian distributions. For the Wasserstein distance, we either use the closed-form Gaussian Wasserstein distance under the same assumption or apply the Sinkhorn algorithm for general distributions. However, in practice, we found that Sinkhorn often fails to converge. We have included this discussion in Appendix D.
>
> | **Method**     | FID ↓      |
> |----------------|----------------|
> | KL             | 23.48          |
> | W-distance     | 18.41          |
> | Sinkhorn       | Not converge   |
> | CS-Aligner     | 12.62          |
>
>
> The results show that our method outperforms the KL and Wasserstein distances. Moreover, Wasserstein distance and KL lack an InfoNCE-style alignment–uniformity decomposition; only CS-divergence yields the compatible formulation (Eq.10). The Gaussian assumption is also stronger than our nonparametric method.
>
> **2. Ablation beyond text-to-image generation.**
>
> The ablation study of each component is reported in Table 7 in Appendix G. For your convenience:
>
> | Method      | FID     |
> |-------------|--------:|
> | InfoNCE     | 151.35 |
> | CS divergence     | 96.72 |
> | CS-Aligner  |  12.62 |
>
> Based on our theoretical analysis, single InfoNCE or CS divergence is insufficient for good alignment. This is because InfoNCE doesn't have distribution alignment ability, while CS divergence lacks pairwise alignment. The integration of the two is critical, as shown in Eq. 10.
>
>
> We perform the ablation between InNCE and InfoNCE+CS divergence (CS-Aligner) on the audio-video retrieval and audio-to-image (A2I) generation tasks. As shown in the table below, CS-Aligner consistently outperforms in the retrieval tasks. This is due to the intra-modality uniformity optimization in the objective function (shown in Eq. 9), which encourages the sample separability (beneficial for the retrieval task). The result for the A2I generation task shows that the InfoNCE-only loss is insufficient to align the two distributions together (crucial for the generation task).
>
> These new results have been added in Appendix H.
>
>
> | **Model**        | **Retrieval** |       |       |       | **Generation** |
> |------------------|---------------|-------|-------|-------|----------------|
> |                  | **V2A**       |       | **A2V** |     | **A2I**        |
> |                  | R@1            | R@5    | R@1   | R@5    | FID            |
> | InfoNCE | 46.1          | 76.9  | 41.2  | 74.4  | 48.19              |
> | CS-Aligner         | 47.7          | 77.2  | 42.2  | 75.3  | 40.06              |
>
>
>
> **3. Clarify how the visualizations in Figure 1.**
>
> For Figures 1a and 1b, we train the same adapter on top of the CLIP model by InfoNCE and CS-Aligner, respectively. We use the MSCOCO training set and visualize the learned representations using t-SNE on 5K image-text pairs from the validation set. For the temperature used in both InfoNCE and CS-Aligner, we resume the temperature from the pretrained CLIP model, and also make it learnable during training.
>
> For Figure 1c, we calculate the L2 distance between the embeddings of all image-text pairs and visualize the histogram. The histogram of the L2 distance between the positive pairs systematically reflects the distribution of distances.
>
> We have added this information in Appendix G.
>
> **4. Qualitative analysis: “present results on other tasks—such as cross-modal retrieval and image captioning”**
>
> For image captioning, we have added qualitative results in Appendix H. The generated captions are semantically aligned with the images, demonstrating the general applicability of our method.
>
>
> **5.  Why does a larger lambda lead to worse results? & The sensitivity of lambda and sigma on other tasks.**
>
> A large lambda will emphasize too much on the distribution alignment, which optimizes the intra-modality uniformity and global distance, while overlooking the pairwise distance. As the generation task is sensitive to both global distribution distance and sample pairwise distance
>
> Based on your suggestion, we perform the sensitivity of $\lambda$ and $\sigma$ on the MSCOCO retrieval task. It shows that the method is robust and performs well with respect to a wide range of hyperparameters in the retrieval task.
>
> We have included these new results and discussions in Appendix H.1.
>
>
> | Retrieval | $\sigma = 0.1$ | 0.5 | 1 | 1.5 |
> |---------------|-------------|---------|-------|---------|
> | R@1     |          49.3   |   50.9      |    50.7   |    50.2     |
>
>
> | Retrieval | $\lambda = 0.01$ | 0.1 | 1 | 10 |
> |---------------|--------------|---------|-------|--------|
> | R@1     |      50.1        |   50.6      |  50.7     |  48.6      |

---

> > ### Author Response · Authors · 2025-11-20
> >
> > **6. Typos.**
> >
> > Thank you for pointing out the typos. We have corrected it in the revision.
> >
> > ---
> >
> > *Thank you for your time and efforts. We hope the provided clarifications have addressed your concerns. Please don’t hesitate to reach out if you have further questions or need more information.*

---

### Official Review · Reviewer_mzxf · 2025-10-30

**Soundness:** 3
**Presentation:** 2
**Contribution:** 2
**Rating:** 4
**Confidence:** 3

**Summary:**

This paper presents CS-Aligner, a novel framework for vision-language alignment that combines Cauchy–Schwarz (CS) divergence with mutual information (InfoNCE). The central idea is to extend alignment beyond individual pairs by introducing distributional alignment, which effectively resolves the inherent tension between the alignment and uniformity objectives in InfoNCE. While InfoNCE encourages matched pairs to come closer (alignment) and all samples to spread apart (uniformity), this dual objective often leads to suboptimal multimodal representation. CS-Aligner addresses this issue by aligning both the pairwise semantics and the global distributions of visual and textual modalities, resulting in more coherent and consistent cross-modal embeddings.

**Strengths:**

1.  Clearly identifies InfoNCE’s limitations—its conflation of alignment and uniformity objectives and dependence on strong negatives that weaken cross-modal consistency.

  2.  Introduces CS divergence as a novel, theoretically grounded approach for distributional alignment, effectively addressing the shortcomings of sample-level contrastive methods.

**Weaknesses:**

1.  The experiments are mainly focused on t2i and i2t tasks, which are not sufficient to verify the effectivess.

  2. The ablation studies in Appendix H.1 are insufficient and unconvincing. The experimental settings are not clearly defined. Furthermore, the paper fails to adequately explain the core concepts of InfoNCE and CS-Aligner, and does not justify the significant performance degradation observed when using only InfoNCE. While these experiments are in the appendix, it is essential to treat them with the same seriousness as those in the main text.

  3. The performance of the CS divergence is sensitive to the choice of kernel and bandwidth, but the paper does not provide a robustness analysis to assess this sensitivity.

**Questions:**

1. It would be better to include more multimodal tasks to verify the effectiveness, as the proposed method is designed for multimodal alignment.

2. Can CS-Aligner generalize to other modality pairs (e.g., audio–text)?

3. Are there scenarios where distributional alignment might unintentionally blur modality-specific distinctions？

---

> ### Author Response · Authors · 2025-11-20
>
> *We sincerely thank #Reviewer mzxf for their insightful comments. The following addresses their concerns and provides answers to their questions.*
>
> ---
>
> **W1 & Q1,2. More experiments.**
>
> To demonstrate the scalability of our method to other multimodal tasks, we extend our method to the video–audio retrieval and generation task. Specifically, we use the VGGSound dataset, randomly selecting 1000 videos for testing and using the rest for training. We sample 4 frames from each video and use the audio to generate 4 images for computing the FID score, which evaluates the audio-to-image generation quality. We compare against ImageBind [1], which is trained on a large-scale dataset using pairwise InfoNCE. We also fine-tune ImageBind using the same training data. Following ImageBind, we also use the fixed temperature, which has been discussed to be beneficial for the audio-image task in ImageBind. Our method consistently outperforms the InfoNCE-only method, ImageBind, on both retrieval and generation tasks. We have included these new results in Appendix H.
>
>
>
> | **Model**        | **Retrieval** |       |       |       | **Generation** |
> |------------------|---------------|-------|-------|-------|----------------|
> |                  | **V2A**       |       | **A2V** |     | **A2I**        |
> |                  | R@1            | R@5    | R@1    | R@5    | FID   ↓      |
> | **ImageBind**    | 21.3          | 44.5  | 20.1  | 43.7  | 53.24         |
> | **ImageBind-finetune** | 46.1          | 76.9  | 41.2  | 74.4  | 48.19              |
> | **Ours**         | 47.7          | 77.2  | 42.2  | 75.3  | 40.06              |
>
>
>
> [1]. Girdhar R, El-Nouby A, Liu Z, et al. Imagebind: One embedding space to bind them all[C]//Proceedings of the IEEE/CVF conference on computer vision and pattern recognition. 2023: 15180-15190.
>
> **W2. Ablation studies clarification.**
>
> (a) *Experimental setting details.* We ablate InfoNCE and CS-Aligner (InfoNCE with CS divergence) on the text-to-image generation task. Specifically, we train the adapter on the MSCOCO dataset and use the Kandinsky decoder to generate the corresponding images. For the temperature in the InfoNCE term, we resume from the pretrained CLIP model and keep it learnable. Then, we compute the FID score for comparison (the lower the better). We have revised this in the revision version.
>
>
> (b) *Justification of the performance degradation using only InfoNCE*.
>
> As clarified in Section 2, InfoNCE contains an inherent alignment–uniformity conflict: the alignment term pulls matched pairs together, while the uniformity term pushes all embeddings apart, including positive pairs. As the learnable temperature $\tau$ (inherited from CLIP) decreases during training, the contrastive logits become sharper, making the uniformity term dominate over the alignment term and thereby weakening multimodal alignment (see our decomposition in Sec. 2) For text-to-image generation, the decoder requires the two modalities to lie in the same distribution, which InfoNCE alone is unable to guarantee. Hence, an InfoNCE-only model may still perform well in cosine-similarity–based retrieval but fails in generation due to the persistent distributional gap.
>
> We also provide new retrieval ablations. Retrieval requires only correct relative similarity ranking, not full distributional overlap, so the degradation of InfoNCE-only is smaller. Nevertheless, CS-Aligner consistently outperforms InfoNCE-only, likely because the intra-modality uniformity terms (Eq. 9) promote better sample separability, which benefits retrieval.
>
>
> | Method     | T2I  | I2T | Avg |
> |----------------|---------|---------|---------|
> | InfoNCE        | 50.1    | 65.8    | 57.95   |
> | CS-Aligner   | 50.7    | 66.34   | 58.52   |
>
>
> **W3. Robustness analysis on KDE to assess the sensitivity.**
>
>
> **Robustness to kernel function**
>
> We choose the Gaussian kernel for its unique theoretical advantage: with it, we can derive the closed‐form compatible formulation (Eq. 10) that unifies InfoNCE’s uniformity and alignment terms. No other kernel admits the conditions required for this derivation. Hence, we consistently use the Gaussian kernel.
>
>
> **Robustness to kernel bandwidth**
>
> In our paper, we analyze the robustness to the kernel bandwidth in two ways. Theoretically, in Appendix E, we justify that with the chosen kernel size $\sigma=1$, the KDE estimation is robust and stable for the high-dimensional space.  Because we L2-normalize embeddings onto the unit hypersphere, pairwise distances concentrate around $\sqrt{2}$​ in high dimensions; using a fixed Gaussian bandwidth $\sigma=1$ keeps kernel values in a well-behaved range, yielding low-variance KDE/CS estimates and stable gradients.
> Empirically, we conduct the sensitivity analysis in Table 7(b), showing that the performance is stable in a certain range of kernel size ($\sigma \approx 0.1-1.5$).

---

> > ### Author Response · Authors · 2025-11-20
> >
> > **Q3. Are there scenarios where distributional alignment might unintentionally blur modality-specific distinctions?**
> >
> > In our distributional alignment, the two distribution masses close while keeping their internal uniformity (Eq. 10), which has a limited influence on the modality-specific distinctions.
> > Moreover, the modality encoders are frozen. CS-Aligner only trains lightweight adapters/LoRA, while the original image and text backbones remain unchanged, preserving modality-specific features.
> >
> >
> > ---
> >
> > *Thank you for your time and efforts. We hope the provided clarifications have addressed your concerns. Please don’t hesitate to reach out if you have further questions or need more information.*
> >
> > ---

---

> > > ### Author Response · Authors · 2025-11-28
> > >
> > > Thank you once again for your thoughtful review and valuable comments! With only five days remaining before the end of the Discussion Phase, we would greatly appreciate your feedback on our revised responses. Please let us know whether our clarifications adequately address your concerns. If any issues remain unresolved or if further explanation is needed, we would be grateful for your additional guidance.
> > >
> > > Best regards, ﻿
> > >
> > > Authors

---

### Official Review · Reviewer_54pL · 2025-10-31

**Soundness:** 2
**Presentation:** 3
**Contribution:** 3
**Rating:** 6
**Confidence:** 4

**Summary:**

The paper introduces CS-Aligner, a novel vision-language alignment framework that combines Cauchy-Schwarz divergence with InfoNCE to address the alignment–uniformity conflict. It enables both global distribution-level and fine-grained token-level alignment, supports unpaired and multi-caption data, and achieves superior performance on generation and retrieval tasks, particularly in low-resource scenarios.

**Strengths:**

1. The authors integrate Cauchy-Schwarz divergence with mutual information (InfoNCE) and provide a theoretical analysis of how it alleviates the alignment–uniformity conflict, addressing limitations in InfoNCE-based alignment.
2. Leveraging distribution-level alignment, the method naturally supports unpaired data and multi-caption supervision, offering greater flexibility and scalability for real-world applications.
3. The use of lightweight adapters (e.g., Adapter, LoRA) enables efficient alignment with CLIP and LLMs, significantly reducing training cost and making the method practical for resource-constrained settings.

**Weaknesses:**

1. Although the method is well-motivated, the theoretical analysis of convergence, generalization, or stability of CS-Aligner remains shallow and largely empirical.
2. The KDE-based CS divergence may introduce additional computational overhead and sensitivity to kernel bandwidth selection, which is not systematically analyzed.
3. Some comparisons (e.g., Eclipse, LLM2CLIP) reuse decoders from other works, which may not fully isolate alignment quality from downstream generative components.
4. The benefits of using unpaired data or token alignment are demonstrated qualitatively, but quantitative ablations are limited and lack statistical rigor.

**Questions:**

1.	How sensitive is the CS divergence estimation to kernel bandwidth (\sigma) and the choice of kernel function?
2.	Does the KDE-based divergence scale efficiently to very large multimodal datasets (e.g., LAION-400M)?
3.	Can the proposed distributional alignment be combined with optimal transport or contrastive regularization for improved interpretability?
4.	How does token-level CS alignment affect semantic disentanglement across layers? Is it stable under random seed variation?
5.	Does the framework ensure mutual consistency between global (distributional) and local (pairwise) alignment objectives?
6.	Could the approach be extended to temporal or 3D multimodal tasks (e.g., video–text, audio–image)?
7.	Given its use of CS divergence, can the model handle multimodal imbalance (e.g., text richer than image features)?

---

> ### Author Response · Authors · 2025-11-20
>
> *We sincerely thank #Reviewer 54pL for their insightful comments. The following addresses their concerns and provides answers to their questions.*
>
> ---
>
> **W1. Theoretical analysis of convergence, generalization, or stability  of CS-Aligner.**
>
> For stability analysis, in Appendix E, we theoretically justify that with the chosen kernel size $\sigma=1$, the KDE estimation is robust and stable in the high-dimensional space. Specifically, as we L2-normalize embeddings onto the unit hypersphere, pairwise distances concentrate around $\sqrt{2}$​ in high dimensions, yielding low-variance KDE/CS estimates and stable gradients. The generalization properties of InfoNCE and CS-divergence–based regularization are discussed in [1] and [2], respectively.
>
> It should be emphasized that a full theoretical analysis of convergence and generalization for both terms is not the goal of this paper. Instead, we aim to provide a theoretical justification for understanding the modality gap in vision-language alignment, as well as a practical solution. To this end, we develop a distributional alignment framework based on Cauchy–Schwarz divergence, and then theoretically show its ability to resolve the alignment–uniformity conflict inherent in InfoNCE.  Indeed, this work is the first to introduce CS divergence into the vision-language alignment area.
> Moreover, our empirical evidence shows that the method converges and generalizes well on zero-shot generation and retrieval tasks.
>
>
>
> **W2. KDE computation and sensitivity analysis**
>
> We analyze the KDE computational complexity in Appendix E. The key conclusion is that for batch size $N$, the computational complexity of our method is $O(N^2)$, which is the same as InfoNCE used in CLIP. This is because both our method and InfoNCE in CLIP are batch-level losses. Thus, our method has comparable computation costs to CLIP.
>
> For the sensitivity analysis, we theoretically justify that with the chosen kernel size $\sigma=1$, the KDE estimation is robust and stable for the high-dimensional space. Empirically, we conduct the sensitivity analysis in Table 7(b), showing that the performance is stable in a certain range of kernel size ($\sigma \approx 0.1-1.5$).
>
>
>
> **W3. Reuse of decoders**
>
> We would like to clarify that we fully isolate alignment quality from downstream generative components by using the same *frozen* decoder as other methods. By doing so, the performance is only affected by the alignment quality.
>
>
> **W4. Quantitative ablations of unpaired data and token alignment**
>
> The quantitative ablations of unpaired data and token alignment are provided in Figure 5 and Figure 4b, respectively, with quantitative FID scores being highlighted. For your convenience, we report the results of two ablations in the following tables. The results show the effectiveness of the token alignment and the use of more unpaired data.
>
> | | FID |
> |----------:|-------------------------------:|
> |   w/o token alignment |                          12.62 |
> |   w/ token alignment |                          12.14|
>
>
>
> |# paired data  | # unpaired data | FID|
> |----------:|-------------------------------:|------------------------------:|
> |  80K |  0   |                        12.62|
> | 40K   |  0   |                          14.71 |
> | 40K  |   40K  |                        12.18|
>
>
>
> **Q1. Sensitivity to the kernel bandwidth and kernel function.**
>
> **Sensitivity to kernel function**
> We choose the Gaussian kernel for its unique theoretical advantage, which is the *only choice* to derive the compatible formulation (Eq. 10) that unifies InfoNCE’s uniformity and alignment terms. Therefore, we only use the Gaussian kernel in our method. We have included this discussion in Appendix H.
>
> **Sensitivity to kernel bandwidth**
> We report the full bandwidth sensitivity analysis in Appendix G. Here, we summarize an ablation on 10 K MS COCO validation samples using FID as the metric. Across a reasonable range of $\sigma$, FID scores exhibit only minor fluctuations, demonstrating that our method is robust to the bandwidth hyperparameter.
>
> | $\sigma$ | 0.1   | 0.5   | 1     | 1.5   |
> |----------|-------|-------|-------|-------|
> |FID      | 30.58 | 27.79 | 29.86 | 31.49 |

---

> > ### Author Response · Authors · 2025-11-20
> >
> > **Q2. Scalability to LAION-400M**
> >
> > The computation cost of our method is comparable to the CLIP-based method when scaling up to even larger-scale datasets. The computational complexity is
> > $O(N^2)$, which is the same as the InfoNCE used in CLIP. Note that N is the batch size, and loss is built upon a batch. Therefore, our method can be scaled efficiently to large-scale datasets like the CLIP-based methods. Moreover, we would like to highlight that our method has been verified in CC12M, a 12-million-scale dataset, showing scalability on the large-scale dataset.
> >
> > As a university-based group, we do not have such computational resources to conduct experiments on the LAION 400M dataset, which requires around 3500 A100 GPU hours for only 10 epochs of training. Given the computational complexity analysis and the success on the CC12M, the method is feasible to scale up to such datasets.
> >
> >
> > **Q3. Combination with OT for improved interpretability.**
> >
> > We are not entirely sure what “interpretability” refers to here. Both OT and our method aim to align distributions. However, the two objectives are not complementary. Since OT does not provide an InfoNCE-style alignment–uniformity decomposition, we use the CS-divergence for the compatible formulation (Eq. 10).
> >
> > To further show the comparison with OT, we also compare the Gaussian-assumed OT and Sinkhorn (general distribution assumption). Sinkhorn fails to converge, and our method outperforms the Gaussian-assumed OT. The comparison with OT has been added in Appendix D.
> >
> > | CS-Aligner | Gaussian-assumed OT | Sinkhorn  OT    |
> > |----------:|-------------------------------:|---------------|
> > |     12.62 |                          18.41 | Not converge  |
> >
> >
> > **Q4. Token alignment: semantic disentanglement across layers & stability under random seed variations.**
> >
> > Our token-level CS alignment builds on top of this CLIP-style representation and focuses on improving fine-grained cross-modal token alignment. Since the CLIP backbones are frozen, we do not change the layer-wise semantic hierarchy and semantic disentanglement across CLIP layers, but rather make high-level token representations more consistently aligned across modalities. Empirically, adding the token-level CS loss improves FID (12.62 to 12.14), suggesting that local visual–textual semantics are better preserved and matched.
> >
> >
> > We train the token alignment experiments with 3 different random seeds, and the performance is stable, as $12.56 \pm 0.86$.
> >
> >
> > **Q5. Mutual consistency of the framework between distributional and pairwise alignment objectives.**
> >
> > Our framework is explicitly designed so that the global (distributional) CS divergence term and the local (pairwise) InfoNCE term remain compatible. The CS divergence drives the overall image and text distributions closer in the embedding space, while the pairwise loss enforces local alignment for individual image–text pairs. When all matched pairs are pulled close, their empirical distributions inevitably become close as well, making the global and local objectives naturally consistent.
> >
> > **Q6. Extension to other tasks: video-text, audio-image.**
> >
> > To show the scalability of our method to other multimodal tasks, we extend our method to the video–audio retrieval and generation task. Specifically, we use the VGGSound dataset, randomly selecting 1000 videos for testing and using the rest for training. We sample 4 frames from each video and use the audio to generate 4 images for computing the FID score, which evaluates the audio-to-image generation quality. We compare against ImageBind [3], which is trained on a large-scale dataset using pairwise InfoNCE. The results in the table below show that our method outperforms the InfoNCE-based method on both generation and retrieval tasks.  We have included this discussion in Appendix H.
> >
> >
> > | Model        | Retrieval |       |       |       | Generation |
> > |------------------|---------------|-------|-------|-------|----------------|
> > |                  | V2A       |       | A2V |     | A2I        |
> > |                  | R@1            | R@5    | R@1    | R@5    | FID    ↓        |
> > | ImageBind    | 21.3          | 44.5  | 20.1  | 43.7  | 53.24         |
> > | ImageBind-finetune | 46.1          | 76.9  | 41.2  | 74.4  | 48.19              |
> > | Ours         | 47.7          | 77.2  | 42.2  | 75.3  | 40.06              |

---

> ### Author Response · Authors · 2025-11-20
>
> **Q7. Multimodal imbalance (e.g., text richer than image features)**
>
>
> Yes, our distributional alignment naturally accommodates multimodal imbalance because it aligns the two modalities at the distribution level using sampled sets. We also validate this empirically. In Fig. 5, we show the multi-caption scenario, where the text modality is richer than the image modality. Incorporating multiple captions leads to improved performance, with the FID score decreasing from 12.62 to 11.78.
>
>
> [1]. Lei Y, Yang T, Ying Y, et al. Generalization analysis for contrastive representation learning[C]//International Conference on Machine Learning. PMLR, 2023: 19200-19227.
>
> [2]. Yin W, Yu S, Lin Y, et al. Domain adaptation with Cauchy-Schwarz divergence[J]. UAI, 2024.
>
> [3]. Girdhar R, El-Nouby A, Liu Z, et al. Imagebind: One embedding space to bind them all[C]//Proceedings of the IEEE/CVF conference on computer vision and pattern recognition. 2023: 15180-15190.
>
> ---
>
> *Thank you for your time and efforts. We hope the provided clarifications have addressed your concerns. Please don’t hesitate to reach out if you have further questions or need more information.*

---

### Official Review · Reviewer_8VHb · 2025-11-01

**Soundness:** 4
**Presentation:** 4
**Contribution:** 4
**Rating:** 10
**Confidence:** 5

**Summary:**

This paper tackles an important but overlooked problem in InfoNCE used in CLIP when aligning text features and image features. The authors revealed that InfoNCE
has inherent conflict in terms of alignment and uniformity in multimodality and demonstrated in Figures 1 and 2 that this is the cause of the modality gaps. Therefore, in this paper, a new method called CS-Aligner was proposed that combines InfoNCE with Cauchy-Schwarz (CS) divergence. Two standard experiments in image-text alignment are done to validate the proposed CS-Aligner: text to image generation and image-text retrieval. In the text to image generation, it has been experimentally demonstrated that the proposed CS-Alignment can achieve better performance than previous large-scale methods and small-scale alignment (Table 1). Its generality has also been validated by using different datasets (Table 2). In the image-text retrieval task, it was shown that CS- Aligner outperforms previous approaches such as Long-CLIP (Zhang et al., 2025) and LLM2CLIP (Huang et al., 2024).

**Strengths:**

The paper is described in detail. The proposed method is mathematically solid. This paper gives important insight to the InfoNCE loss and proposes an important method that would give great impacts to the multimodal research community.

Experimental results both in the main body and supplementary are convincing enough.

**Weaknesses:**

The proposed method has some parameter sensitivity as shown in Table 7 in the supplemental material. I do not think the method is parameter-independent as the authors claim.
- Could you show soma sample images to show the method is robust to parameter changes, in addition to showing FID?
- Is there any method that can automatically find optimal parameter settings rather than relying on users’ empirical optimization?

Some minor modification proposals (no need to reply)
- A missing period after eq. (21)
- A missing period, a missing eq. number, and erroneous “?” in lines 911-913
- “??” in line 1065 should be modified

**Questions:**

None

---

> ### Author Response · Authors · 2025-11-20
>
> *We sincerely thank #Reviewer 8VHb for their insightful comments. The following addresses their concerns and provides answers to their questions.*
>
> ---
>
> **1. Parameter sensitivity.**
>
> We apologize for the confusion. By “non-parametric KDE”, we meant no parametric distributional form, not “parameter-independent”.
>
> In statistics, KDE is non-parametric in the sense that it does not assume a fixed finite-dimensional parametric family; however, it involves a small number of hyperparameters (e.g., bandwidth) whose values can influence performance.
>
> Our method indeed has standard training hyperparameters (e.g., the Gaussian kernel bandwidth $\sigma$), for which we report sensitivity in Table 7; results show robustness within a certain range.
>
> We have clarified this in line 215 to make it clearer.
>
>
> **2. Sample images with respect to parameter change.**
>
> Thank you for the suggestion. We have added the qualitative comparison with respect to different kernel widths in Appendix H. From the quantitative results, one can obtain a similar conclusion: the method is robust to a certain range of kernel width changes.
>
>
> **3. Automatic parameter determination strategy.**
>
> There exist alternative strategies to set the kernel bandwidth, e.g., heuristic choices such as the batch-mean/median distance (the “median heuristic”) [1].
>
> We did not adopt these for stability reasons. Heuristic bandwidths such as batch mean can be noticeably affected by batch-level outliers (especially with small or heavy-tailed batches), leading to unstable and drifting $\sigma$. Although the batch median is more robust to such outliers than the mean, we observed that the training is still not stable in deep neural networks. Our choice of L2-normalized embeddings with a fixed global $\sigma$ is more robust to outliers and avoids injecting batch noise into the objective. Also, because we L2-normalize embeddings onto the unit hypersphere, pairwise distances concentrate around $\sqrt{2}$​ in high dimensions; using a fixed Gaussian bandwidth $\sigma=1$ keeps kernel values in a well-behaved range, yielding low-variance KDE/CS estimates and stable gradients.
>
>
>
>
> **4. Typos.**
>
> Thank you for pointing out the typos. We have corrected them in the revision.
>
> —
>
> [1]. Gretton A, Borgwardt K M, Rasch M J, et al. A kernel two-sample test[J]. The journal of machine learning research, 2012, 13(1): 723-773.
>
> ---
>
> *Thank you for your time and efforts. We hope the provided clarifications have addressed your concerns. Please don’t hesitate to reach out if you have further questions or need more information.*

---

### Meta-Review · Area_Chair_4Htg · 2026-01-05

**Summary:**

- Theoretical analysis of convergence, generalization, or stability of CS-Aligner remains shallow and largely empirical.

- Additional computational overhead and sensitivity to kernel bandwidth selection.

- Quantitative ablations are limited and lack statistical rigor.

- Lacks experiments, the ablation studies in Appendix H.1 are insufficient and unconvincing.

- The performance of the CS divergence is sensitive to the choice of kernel and bandwidth, but the paper does not provide a robustness analysis to assess this sensitivity.

- Lacks quantitative comparisons with other distribution distance measures.

- Clarity.

**Reviewer Concerns:**

I think the concerns about the theoretical analysis and sensitivity still exist. The authors have added some results to respond to the questions about the missing experiments.

**Reviewer Scores:**

I think Reviewer mzxf might slightly increase the score.

---

### Decision · Program_Chairs · 2026-01-26

Accept (Poster)